# Revisiting the role of oxidation in stable and high-performance lead-free perovskite-IGZO junction field-effect transistors

Seonkwon Kim[1], Su Hyun Kim [2], Hui Ung Hwang[3,4], Jeongmin Kim[5], Jeong Won Kim [3,4], In Cheol Kwak [1], Byeongjae Kang [6], Seungjae Lee[1], Sae Byeok Jo [7,8], Du Yeol Ryu [1], Hyunjung Kim [6], Jae-Min Myoung [9], Moon Sung Kang [10,11] ✉, Saeroonter Oh [12] ✉ & Jeong Ho Cho [1] ✉

Mitigating the oxidation susceptibility of Sn remains a critical issue for improving the environmental stability of lead-free perovskites. Herein, we show that the oxidized surface layer of Sn-based perovskites can be utilized to improve transistor performance, rather than being entirely suppressed. We report perovskite-IGZO junction field-effect transistors that use this oxidized layer to suppress gate current to below $10^{-10}$ A, enabling enhancement-mode operation. We refer to these as barriered junction field-effect transistors. The combination of the gate leakage suppression and high polarizability of the perovskite layer results in a field-effect mobility of 29.4 $cm^2V^{-1}s^{-1}$, subthreshold swing of 67.1 mV $dec^{-1}$, and on/off current ratio exceeding $10^5$ under ≤1 V operation. These devices maintain stable operation in ambient conditions. Furthermore, we demonstrate their applicability by constructing logic gates such as NOT, NOR and NAND. These findings highlight the potential of exploiting Sn-based perovskite oxidation to advance electronic devices.

Organometallic halide perovskites have gained significant attention owing to their tunable bandgap, facile processability, and high carrier mobility[1–3]. These properties have enabled major advances in optoelectronics, particularly in solar cells and light-emitting diodes, while also opening new possibilities for transistor applications[4–11]. Although Pb-based perovskites have demonstrated the most excellent optoelectronic properties, their toxicity poses serious environmental concerns and limits industrial applications[12–14]. Sn-based perovskites are considered attractive substitutes owing to their valence electron configuration, which resembles that of their Pb-based counterparts[11,15–17]. However, the susceptibility of Sn to oxidation due to the absence of the lanthanide shrinkage effect, transitioning from $Sn^{2+}$ to $Sn^{4+}$, has hindered their practical implementation[18–20]. Previous research has shown that Sn-based perovskites demonstrate highly promising performance as the p-type channel material in thin-film transistors; however, even the slightest exposure to air within mere minutes leads to self-p-doping, which results in metallic characteristics that hinder their function as transistors[4,11,21–25]. While numerous

[1]Department of Chemical and Biomolecular Engineering, Yonsei University, Seoul, Republic of Korea. [2]Department of Electrical and Computer Engineering, Sungkyunkwan University, Suwon, Republic of Korea. [3]Korea Research Institute of Standards and Science (KRISS), Daejeon, Republic of Korea. [4]University of Science and Technology (UST), Daejeon, Republic of Korea. [5]Division of Nanotechnology, DGIST, Daegu, Republic of Korea. [6]Department of Physics, Center for Ultrafast Phase Transformation, Sogang University, Seoul, Republic of Korea. [7]School of Chemical Engineering, Sungkyunkwan University (SKKU), Suwon, Republic of Korea. [8]SKKU Institute of Energy Science and Technology (SIEST), Sungkyunkwan University, Suwon, Republic of Korea. [9]Department of Materials Science and Engineering, Yonsei University, Seoul, Republic of Korea. [10]Department of Chemical and Biomolecular Engineering, Sogang University, Seoul, Republic of Korea. [11]Institute of Emergent Materials, Ricci Institute of Basic Science, Sogang University, Seoul, Republic of Korea. [12]Department of Semiconductor Convergence Engineering, Sungkyunkwan University, Suwon, Republic of Korea. ✉e-mail: kangms@sogang.ac.kr; sroonter@skku.edu; jhcho94@yonsei.ac.kr

research efforts have focused on suppressing this oxidation through various chemical and physical strategies, these approaches face inherent limitations[26,27]. Considering the spontaneity of Sn oxidation and distinctive evolution in its electronic structures, the resulting electronic properties of oxidized Sn-based perovskites could potentially open up opportunities in electronic devices.

The increasing prevalence of Internet of Things devices has made low power consumption a critical requirement in modern electronics. Junction field-effect transistors (JFETs) are considered suitable for such applications owing to their steep subthreshold swing (SS) values[28,29]. Unlike conventional metal-oxide-semiconductor field-effect transistors (MOSFETs), which often require advanced techniques (such as vacuum-deposited ultra-thin high-k gate dielectrics) and complex interface engineering[30–42] to achieve optimal performance, JFETs can theoretically achieve near-ideal SS without a gate dielectric layer owing to their inherently high capacitance[29]. However, this architecture presents two significant challenges: substantial gate leakage current and operation limited to depletion mode. The depletion-mode constraint is particularly problematic for power consumption since these devices conduct significant current at zero gate voltage. Furthermore, achieving enhancement-mode operation in conventional JFETs is fundamentally limited by large forward bias currents at the PN junction[43,44]. Due to these inherent limitations, research into JFET technologies has made limited advancement in recent years, while alternative transistor architectures such as MOSFET and high electron mobility transistors (HEMTs) have gained increasing prominence. Yet, these enhancement-mode operating devices are realized at the expense of increased fabrication complexity. For instance, MOSFETs often require gate stack dipole engineering, while HEMTs typically require epitaxially grown heterostructures and precise barrier/channel band alignment strategies[45]. Therefore, developing JFETs capable of enhancement-mode operation while maintaining low leakage current remains a crucial challenge for advancing low-power electronics.

In this study, we report a simple, low-temperature, solution-processable, scalable strategy to fabricate high-performance enhancement-mode JFETs by tailoring and leveraging the oxidation phenomenon of Sn-based perovskites. Specifically, a PN junction is formed using p-type $PEA_2SnI_4$ and n-type indium gallium zinc oxide (IGZO). While the as-fabricated device suffers from large gate leakage, we turn the above-mentioned conventional drawback of Sn-based perovskites (i.e., their susceptibility to oxidation) into a functional advantage. Controlled surface oxidation of the Sn-based perovskite results in the formation of a barrier layer that effectively suppresses gate leakage to below $10^{-10}$ A. This enables enhancement-mode operation, overcoming a key limitation of conventional JFETs. We refer to this architecture as a barriered JFET (b-JFET). Our b-JFET achieves excellent electrical properties, including a field-effect mobility of 29.4 $cm^2V^{-1}s^{-1}$, a low SS of 67.1 mV $dec^{-1}$, and an on/off current ratio exceeding $10^5$ under ≤1 V operation. Moreover, we demonstrate robust environmental, bias, and operational stability over extended periods in ambient air, despite containing a typically unstable Sn-perovskite layer, highlighting the robustness of our approach. Additionally, we successfully constructed logic circuits and achieved high inverter gain values with low applied voltages, further confirming the potential of the device for various practical applications.

## Results and discussion
### Operation mechanism and performance
Figure 1a shows the schematic of the perovskite b-JFETs fabricated using Sn-based perovskite ($PEA_2SnI_4$) as the p-type semiconductor and IGZO as the n-type semiconductor. Device fabrication began with sputtering ITO source and drain electrodes onto the substrate, which were subsequently patterned using conventional photolithography and etched with hydrochloric acid. Thereafter, the IGZO layer was sputtered, patterned through conventional photolithography, and

chemically etched. Parylene-c was then deposited onto the IGZO layer using chemical vapor deposition (CVD) and subsequently patterned via conventional photolithography and reactive ion etching to serve as a hard mask for the $PEA_2SnI_4$ layer. Next, $PEA_2SnI_4$ was spin-coated onto the patterned parylene-c layer, and then a mechanical peel-off process was performed[46,47]. Subsequently, the $PEA_2SnI_4$ layer was subjected to thermal treatment at 60 °C with ~40% relative humidity. The thermal treatment resulted in the formation of a thin oxidized surface layer on $PEA_2SnI_4$, which eventually played a crucial role in the operation of the device. Finally, Au was thermally deposited onto the perovskite layer through a shadow mask to complete the device structure. The overall fabrication process is illustrated in Supplementary Fig. 1, and the optical microscopy (OM) image is shown in Supplementary Fig. 2.

Prior to delving into the operation of the perovskite b-JFET, we first discuss the operation of the device without an oxidized surface layer on $PEA_2SnI_4$. This is to highlight the critical role of the oxidized surface layer. The given device with no oxidized surface layer is a conventional JFET made of a p-type $PEA_2SnI_4$ and an n-type IGZO heterojunction. The resulting device is an n-channel JFET, and its channel current is modulated by controlling the width of the depletion layer at the PN-interface through the application of a reverse bias applied at the gate electrode (Fig. 1b). A typical n-channel JFET operates as a normally-on-state device, where the off-state is obtained when the width of the depletion layer becomes larger than the n-channel thickness by applying a highly negative reverse bias to the gate. Note that a positive gate bias cannot be applied to the n-channel JFET, in principle, as the resulting forward bias applied to the PN junction leads to a dramatic leakage current through the gate (Fig. 1c). Our n-channel JFET not only experienced large leakage at the positive gate bias, but it also suffered from a large leakage current even at the negative gate bias (unlike typical JFETs) because the reverse bias current for our PN heterojunction was significantly large (Supplementary Fig. 3). Specifically, under negative gate voltage ($V_G$), the drain current ($I_D$) was significantly high and closely matched the gate current ($I_G$), indicating that the current originated from gate-drain leakage (further details in the later section). This implies that the reverse-bias current of the PN junction between the gate and drain dominated the device behavior. Overall, although we have successfully fabricated a stack of PN heterojunctions connected to the three electrodes, the resulting devices operated poorly as an n-channel JFET.

Dramatic changes in the current-voltage characteristics were obtained, as the device contained an oxidized surface layer on $PEA_2SnI_4$, which was formed by thermal treatment of the $PEA_2SnI_4$ layer to ambient air (at 60 °C and ~40% relative humidity) during the fabrication process. Notably, the gate leakage current was drastically suppressed (Fig. 1d). In fact, it was discovered that a surface layer comprising $SnO_2$ and PEAI was formed (details in a later section). This layer resulted in a type-II staggered band alignment with $PEA_2SnI_4$. The resulting energy band structure had a large energy barrier that prevented electrons from leaking to the gate, which resulted in a completely different operation scheme for the device. Figure 2a shows an image of the four-inch wafer-scale fabrication of b-JFETs. Although the details of the device operation will be further corroborated with computational simulations in the later section, a brief explanation can be provided as follows. At negative $V_G$, both the $I_D$ and source current $I_S$ can be drastically suppressed (Fig. 2b). This is not only because the width of the depletion layer at the heterojunction interface widens (which should suppress the electron flow between the source and drain) but also because the leakage of electrons from the gate to the source or drain is blocked. Consequently, the genuine off-state of the transistor can be realized at a current level below $10^{-10}$ A, which was the minimum current level that could be recorded with our measurement setup. Recall that the off-state current of the device without the

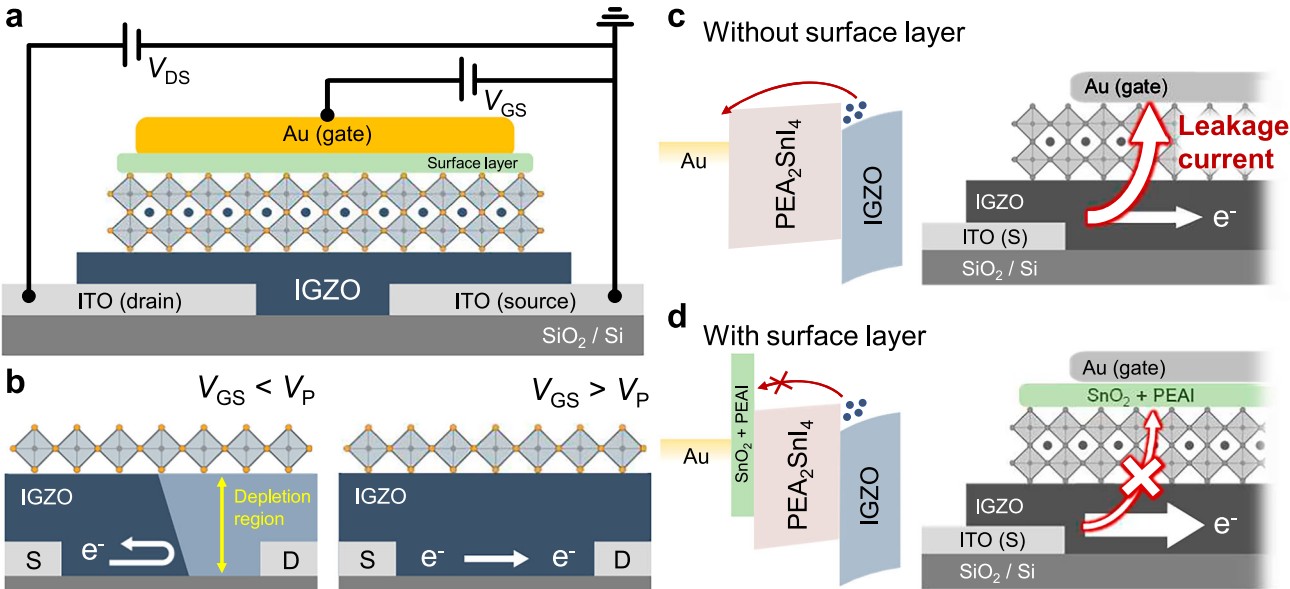

**Fig. 1 | Operation mechanism of perovskite b-JFETs. a, b** Schematic of the perovskite b-JFET based on a PEA$_2$SnI$_4$/IGZO heterojunction and the schematic illustration of the working mechanism. **c, d** Schematic illustration of the electron flow within the device and the corresponding energy band diagram of the perovskite b-JFETs before and after the oxidation process.

surface oxidized layer was ~10$^{-5}$ A, which matched the gate leakage current level. More interestingly, at positive $V_{GS}$, the significant gate leakage (which should be significant for conventional JFETs because their PN junction is forwardly biased) could be suppressed because of the effective electron barrier formed by the oxidized surface layer (the gate current [$I_G$] remains low at positive $V_{GS}$). Consequently, both $I_D$ and $I_S$ could be modulated effectively with positive $V_{GS}$, which cannot be achieved in conventional JFETs and even in our devices without the oxidized surface layer. Overall, the channel current in our devices could be effectively modulated with negative and positive $V_{GS}$ without severe leakage. Unlike conventional MOSFETs, where the gate oxide serves as a gate dielectric that modulates charge carrier density via capacitive coupling, the oxidized surface layer in our b-JFET functions solely as an electron-blocking barrier, controlling junction conduction without invoking charge accumulation mechanisms. The resulting device characteristics at low drain voltages showed comparable performance to those at high drain voltages with an on/off-state current ratio above 10$^5$ and an electron mobility ($\mu$) of 29.4 cm$^2$V$^{-1}$s$^{-1}$ at low voltages of ≤1 V (transfer characteristics measured at different drain voltages are presented in Supplementary Fig. 4). The field-effect mobility ($\mu$) of our devices was calculated using the equation $\mu = (L/W)(g_m)/(q \cdot N_s \cdot t)$, where $L/W$, $g_m$, $q$, $N_s$, and $t$ denote the channel length-to-width ratio, maximum transconductance defined as $\partial I_D/\partial V_G$, element charge, electron density estimated from a separate Hall measurement (Supplementary Fig. 5), and channel thickness, respectively[29,48]. Note that the mobility calculation does not rely on the capacitance measurements, as we employ the JFET mobility calculation equation rather than the conventional MOSFET equation. This classification of our device as a JFET is supported by our impedance analysis (Supplementary Fig. 6), which shows that neither the perovskite layer nor the perovskite-oxide layer functions as a conventional dielectric. The output characteristics of the devices at various $V_{GS}$ values are depicted in Fig. 2c, which shows a channel pinch-off as $V_D$ reaches $V_{GS}-V_T$ (where $V_T$ is the threshold voltage) and the current saturation above this voltage. The operation of our devices is enabled by the critical role of the oxidation step of the perovskite layer, which has been typically considered as a detrimental feature of the materials that must be suppressed. We transformed the disadvantage of the perovskite layer

(oxidation) into a functional advantage critical to enabling the device operation.

Notably, our perovskite b-JFETs exhibited a low SS of 67.1 mV dec$^{-1}$, approaching the theoretical limit of ~60 mV dec$^{-1}$ at 300 K. This result can be attributed to the high intrinsic capacitance ($C_I$) of the SnO$_2$:PEAI and PEA$_2$SnI$_4$ layers located on top of the channel (10 μF cm$^{-2}$ at 1 kHz). These high capacitance values agree with the high capacitance of ionic dielectrics, including the highly polarizable perovskite layers reported in previous research[49]. Theoretically, the SS can be expressed as SS $= \ln(10)mk_BT/q$, where $m$ is the ideality factor, $k_B$ is the Boltzmann constant, $T$ is the temperature, and $q$ is the element charge. Under ideal operations, $m$ is unity, and thus, the SS reaches the Boltzmann limit of 60 mV dec$^{-1}$ at 300 K. However, in practical devices, channel capacitance ($C_s$) also exists and thus the ideality factor deviates from unity following the relation $m = 1 + C_S/C_I$. This relation indicates that the gate stack must have a high $C_I$—similar to our perovskite gate stack—to realize ideal SS operation. We discovered that $m$ was 1.1 using our achieved SS of 67.1 mV dec$^{-1}$. Considering that the potential applied by the gate bias is dropped across the gate stack and the semiconductor, $\Delta V_{GS} = \Delta V_I + \Delta \psi_S$, where $\Delta V_I$ denotes the potential drop on the gate stack, and $\Delta \psi_S$ denotes the change in surface potential in the semiconductor, the given $m$ value implies that the dominant fraction of the gate potential (89%) is transferred to the semiconductor channel to effectively modulate the current density. Furthermore, the low interface trap density at the perovskite/IGZO junction likely contributes to the steep SS observed. Recent studies have shown that perovskite/inorganic oxide buried interfaces, such as those with SnO$_2$, exhibit reduced trap-assisted recombination and efficient carrier extraction, which supports the favorable interfacial electronic properties of such heterojunctions[50].

Figure 2d shows the 2D mapping images of the SS, threshold voltage, and $\mu$ values derived from the transfer characteristics of 100 perovskite b-JFETs in an array. The transfer characteristics of the 100 devices are presented in Supplementary Fig. 7. Figure 2e depicts the air stability of the perovskite b-JFET encapsulated with parylene-c. The air stability of the devices with and without polydimethylsiloxane encapsulation is presented in Supplementary Fig. 8. For the stability measurement, the perovskite b-JFET was stored in ambient air in the

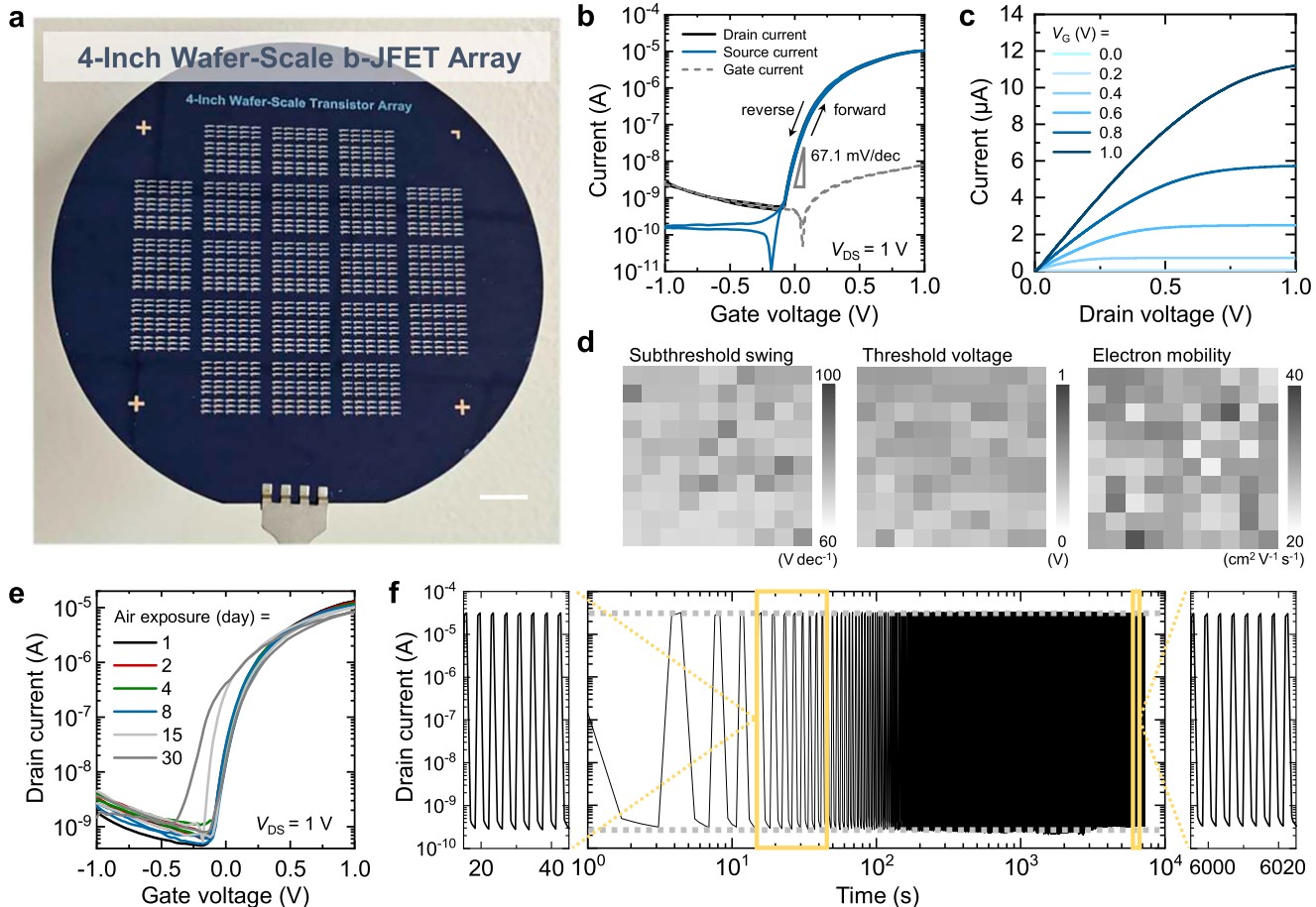

**Fig. 2 | Electrical performances of perovskite b-JFETs. a** Image of the four-inch wafer scale of the perovskite b-JFETs (scale bar = 1 cm). **b** Transfer characteristics of the optimized perovskite b-JFET. **c** Output characteristics of the optimized perovskite b-JFET. **d** 2D Color mapping of the subthreshold swing, threshold voltage, and effective field-effect mobility distribution. **e** Air stability of the optimized perovskite b-JFET. **f** Operational stability measurement of the perovskite b-JFET.

dark at 20 °C and a relative humidity of 40%. Remarkably, the perovskite b-JFET exhibited only a minimal deviation in its transfer characteristics over 1 month. We attribute the excellent air stability of the device to the retarded oxidative reaction kinetics caused by the already-formed protective surface layer acting as a barrier against further penetration of oxygen molecules into the bulk. Ultraviolet (UV)–visible spectroscopy analyses were conducted to investigate the reaction kinetics, as shown in Supplementary Fig. 9. The intensity of the main absorption peak at 610 nm originating from $PEA_2SnI_4$ was investigated as a function of air exposure time. The intensity of the main absorption peak at 610 nm, which originated from $PEA_2SnI_4$, was investigated as a function of air exposure time. Three different samples were compared: pristine $PEA_2SnI_4$ (Pe0) without encapsulation, $PEA_2SnI_4$ annealed in the air for 120 min (Pe120) without encapsulation, and Pe120 with parylene-c encapsulation. The decay of the absorption peak of Pe0 was two times faster than that of Pe120, while the decay rate of the encapsulated Pe120 was significantly reduced. The same trend was also observed in time-dependent photoluminescence analysis as shown in Supplementary Fig. 10. Figure 2f shows the operational stability of the perovskite b-JFET evaluated by repetitively turning the device on and off with a gate bias of 0.5 V and −0.5 V, respectively, for over 7000 s at a frequency of 0.54 Hz. No evident change was observed during the operational stability evaluation, and the on/off current ratio was maintained over $10^5$ during the entire test. The perovskite b-JFETs also exhibited excellent stability in both the bias stress and cyclic stability tests, as shown in Supplementary Figs. 11 and 12.

## Characterization of the oxidized surface layer

To understand the influence of thermal treatment of the $PEA_2SnI_4$ layer, we examined the chemical composition of the surface layers of Pe0, Pe60, Pe90, and Pe120 using X-ray photoelectron spectroscopy (XPS) depth profile analysis upon etching with 500 eV monoatomic Ar ions (Fig. 3a and Supplementary Fig. 13 for Pe60 and Pe90). In particular, the evolution of the Sn 3d and O 1s peaks at different depths of the films was examined. These peaks were deconvoluted to determine the proportion of Sn and O oxidation states in the films. The deconvoluted Sn 3d peaks centered at 487.4 eV and 495.8 eV correspond to $Sn^{4+}$, which indicates the oxidation state of perovskite, whereas those peaks centered at 486.4 eV and 494.8 eV are assigned to $Sn^{2+}$, which is typical of (unoxidized) $PEA_2SnI_4$[20]. The presence of $Sn^{4+}$ peaks in the Pe120 spectra suggests the distinct formation of $SnO_2$ on the surface. As the etching process progressed, the proportion of $Sn^{4+}$ relative to $Sn^{2+}$ decreased from the initial 75% to 11%, which is comparable to that observed in the Pe0 sample. Meanwhile, the O 1s spectra of Pe120 exhibited peaks associated with $SnO_2$, which also decreased concomitantly as the etching process progressed. Figure 3b, c summarizes the proportion of $Sn^{4+}$ and $Sn^{2+}$ and the deconvoluted $O_2$ area. These results imply that the formation of the $SnO_2$-containing surface layer induced by the thermal treatment of the perovskites occurs at a sub-10-nanometer scale. Moreover, ellipsometry measurements of Pe0, Pe60, Pe90, and Pe120 films were conducted to further elucidate the progression of the bilayer structure, confirming the formation and growth of the surface layer as annealing time increased (Supplementary Table 1 and Supplementary Fig. 14). We also conducted X-ray

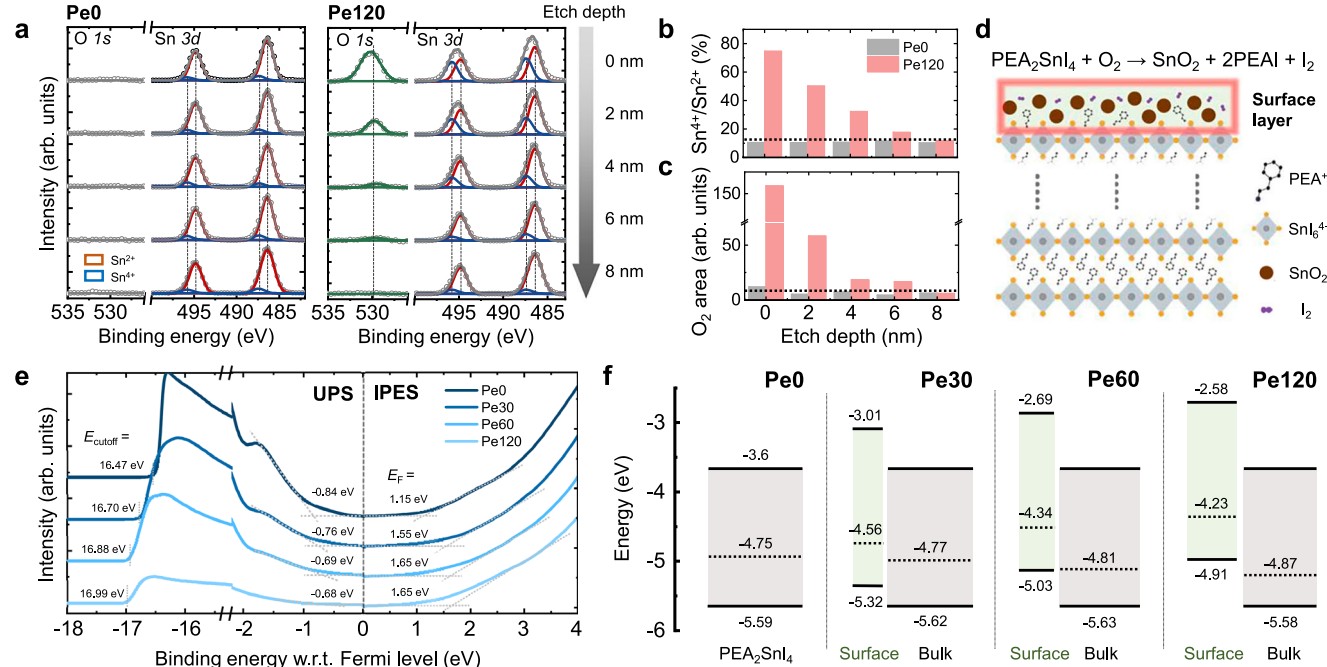

**Fig. 3 | Characterization of the oxidized surface layer. a** XPS depth profiles of Pe0 and Pe120. **b, c** Sn$^{4+}$-to-Sn$^{2+}$ ratios of Pe0 and Pe120 calculated from the deconvoluted peaks of the Sn 3d XPS spectra and the O area of Pe0 and Pe120. **d** Schematic illustration of the chemical reaction upon PEA$_2$SnI$_4$ oxidation. **e** Combined UPS and IPES analysis results of the PEA$_2$SnI$_4$ films with different annealing times. **f** Energy band diagrams of the surface and bulk PEA$_2$SnI$_4$ thin films annealed for various durations.

diffraction (XRD) analysis on these samples (Supplementary Fig. 15) to further elucidate the surface layer. In the XRD spectra of Pe60 and Pe120, the newly generated peaks highlighted in gray are assigned to PEAI[26], which further indicates the formation of the surface layer. Additionally, grazing incidence XRD measurements were conducted on these films (Supplementary Fig. 16), also revealing the progressive growth of the surface layer.

Figure 3d illustrates the overall chemical reactions and the formation of the surface layer. Theoretical and experimental results from a previous study have proposed that exposing PEA$_2$SnI$_4$ to high concentrations of O$_2$ leads to its undergoing multiple-step reactions, ultimately producing various byproducts. In accordance with the previous study, we used various chemical analyses to experimentally demonstrate that the following reaction occurs at the surface when PEA$_2$SnI$_4$ is thermally treated in ambient air: PEA$_2$SnI$_4$ + O$_2$ → SnO$_2$ + 2PEAI + I$_2$. Furthermore, electrochemical impedance spectroscopy (EIS) measurements were conducted to confirm the presence of a surface layer on the device, which had a structure of ITO/IGZO/PEA$_2$SnI$_4$/Au. The resulting Nyquist plot, which features characteristic semicircles, is shown in Supplementary Fig. 17 along with an inset of an equivalent circuit. Notably, the enlarged semicircle in the low-frequency region indicates an increased resistance between IGZO and the perovskite, possibly due to the surface layer formed through oxidation[51]. To scrutinize the role of this surface layer, we compared the transfer characteristics with a device incorporating a thin PMMA layer designed to serve a similar function as the surface oxidation layer (Supplementary Fig. 18). While the PMMA-based devices could be turned on, they exhibited higher leakage current and degraded SS due to the lower dielectric constant of PMMA compared to the surface layer. More significantly, we observed large hysteresis in the transfer curve, which can be attributed to interface traps arising from the heterogeneous junction between PMMA and the perovskite layer. In contrast, our surface layer forms a homogeneous junction with the perovskite, minimizing interface traps and resulting in significantly reduced hysteresis in the transfer characteristics.

Next, ultraviolet photoelectron spectroscopy (UPS) and inverse photoelectron spectroscopy (IPES) were employed to characterize the electronic structures of the perovskite films annealed in ambient air for different durations, including their work function ($E_{WF}$), conduction band minimum (CBM), and valence band maximum (VBM). Figure 3e shows the UPS and IPES analysis results of the pristine PEA$_2$SnI$_4$ thin film (denoted as Pe0) and the PEA$_2$SnI$_4$ thin films annealed in ambient air for 30 min, 60 min, and 120 min (denoted as Pe30, Pe60, and Pe120, respectively). Note that thermal annealing was conducted at 60 °C to simply accelerate the oxidation process. According to the UPS analysis, the VBM of PEA$_2$SnI$_4$ increased gradually as oxidation progressed. The VBM was 5.59 eV for Pe0, but it increased to 4.91 eV for Pe120. The $E_{WF}$ of the films was estimated using the relation $E_{WF} = 21.2$ eV$-E_{cutoff}$, where $E_{cutoff}$ is the secondary electron cutoff obtained from the UPS analysis. The $E_{WF}$ values of Pe0, Pe30, Pe60, and Pe120 were 4.75, 4.56, 4.34, and 4.23 eV, respectively. Meanwhile, the CBM obtained using IPES shifted from 3.6 eV for Pe0 to 3.01, 2.69, and 2.58 eV for Pe30, Pe60, and Pe120, respectively. Note that the above UPS and IPES analysis results reflect the electronic structures of the surface layer. To determine the structures of the bulk layer beneath the oxidized surface layer, the samples were subjected to Ar sputtering to etch their surface layers, followed by UPS measurements (Supplementary Fig. 19). The results revealed that despite the oxidation of the top surface layer, the electronic structure of the bulk remained more or less similar to that of Pe0, with only a small shift in $E_{WF}$ toward the VBM due to oxidation. To electrically verify the energy barriers determined by UPS, we fabricated vertical Au-Pe0-IGZO-ITO and Au-Pe120-IGZO-ITO Schottky diodes and extracted barrier heights through temperature-dependent $I-V$ measurements (Supplementary Fig. 20). Figure 3f depicts the overall energy band diagram based on the obtained values.

**Technology, computer-aided design simulation**

Figure 4a presents the experimental transfer curves of the perovskite b-JFETs based on Pe0, Pe30, Pe60, Pe90, and Pe120, while Supplementary Fig. 21 presents those of the Pe150-, Pe180-, and Pe210-based

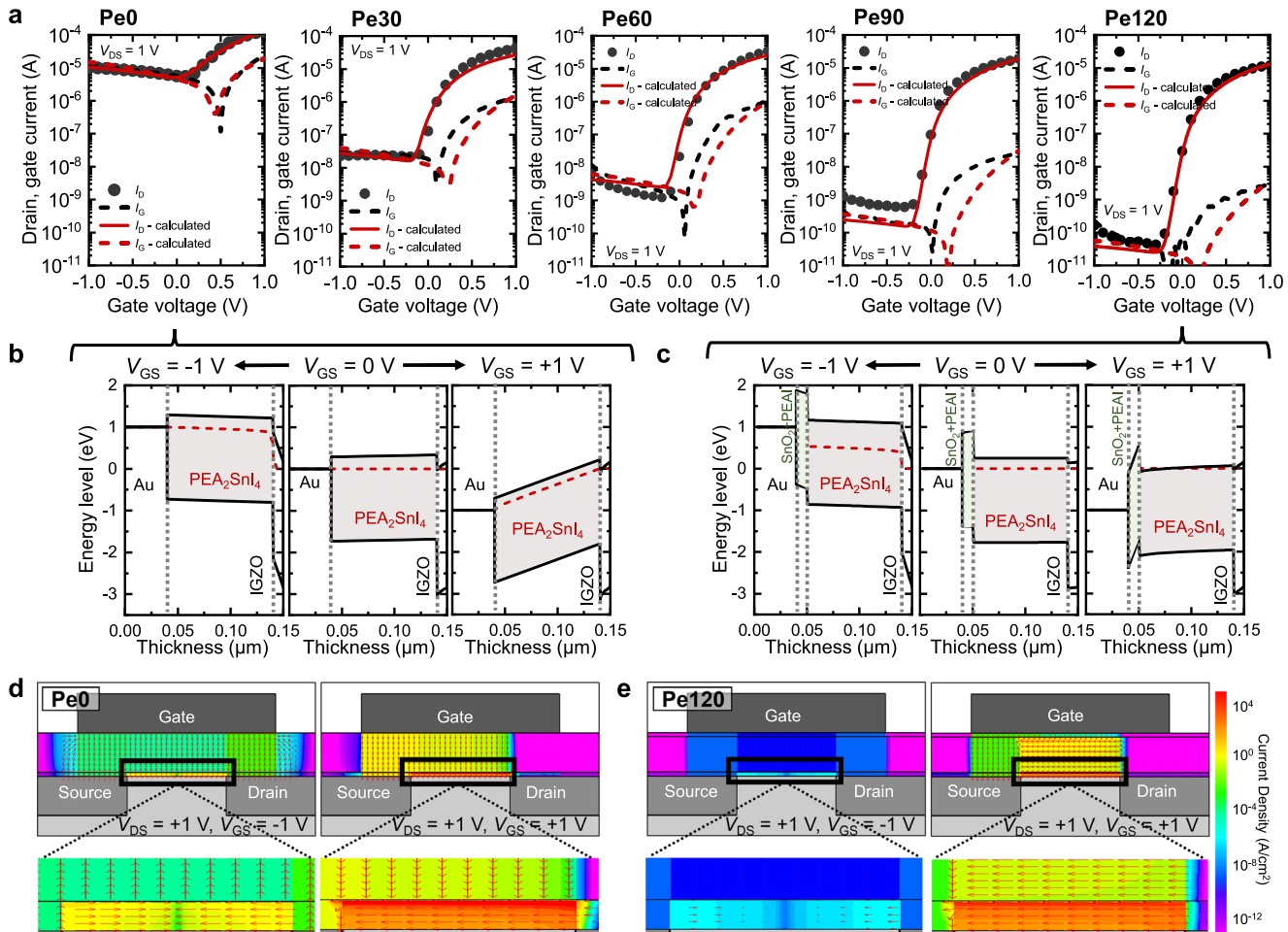

**Fig. 4 | TCAD simulations. a** Experimental and simulated transfer curves of perovskite b-JFETs with different annealing times. Energy band diagrams of the **b** Pe0 and **c** Pe120-based perovskite b-JFETs at different $V_{GS}$ levels. TCAD simulation revealing 2D contour maps of the current density distribution in the **d** Pe0- and **e** Pe120-based perovskite b-JFETs at $V_{GS}$ of −1 V (left panel) and 1 V (right panel).

perovskite b-JFETs. The transfer curves show an apparent decrease in the off-state $I_D$ and $I_G$ as the annealing time of the perovskite increases. The transfer characteristics of perovskite b-JFETs as a function of annealing temperature are presented in Supplementary Fig. 22. Notably, the $I_G$ in the negative $V_{GS}$ range decreased from -$10^{-5}$ A to -$10^{-10}$ A, and the $I_G$ in the positive $V_{GS}$ range decreased from -$10^{-5}$ A to -$10^{-9}$ A. The $I_D$ on/off ratio dramatically increased from $2.4 \times 10^1$ to $2.8 \times 10^5$ because the off-state $I_D$ was primarily due to $I_G$. To gain insight into the relationship between surface layer formation progress and the observed trends, we conducted technology computer-aided design (TCAD) device simulations using Silvaco 2D Atlas. The device structure depicted in Fig. 1a and the experimentally acquired band structure shown in Fig. 3f were employed in the simulation. The conduction band discontinuity ($\Delta E_C$) between PEA$_2$SnI$_4$ and IGZO, as well as the height of the Schottky barrier at the gate, was adjusted to match both the $I_D$ and $I_G$ from the experimental curves. Figure 4a shows that the simulated $I_D$ and $I_G$ curves are in good agreement with the experimental results, confirming the critical role of the surface layer generated from the oxidation of PEA$_2$SnI$_4$ and providing support for the simulation parameters.

The effect of the surface layer on the electrostatic properties and current density distribution was further studied based on the validated simulation results. Figure 4b, c shows the energy band diagrams of the perovskite b-JFETs based on Pe0 and Pe120, respectively, under different gate bias conditions. The energy band diagrams of Pe30, Pe60,

and Pe90 are shown in Supplementary Fig. 23a. At $V_{GS} = +1$ V (a bias condition that causes electrons to flow from the channel to the gate), the electrons in Pe0 only encountered the energy barrier between PEA$_2$SnI$_4$ and IGZO; however, for those in Pe120, the surface layer formed an additional barrier between PEAI+SnO$_2$ and PEA$_2$SnI$_4$, dramatically suppressing the leakage of electrons to the gate. At $V_{GS} = −1$ V (a bias condition that causes electrons to flow from the gate to the channel), the thermionic emission of electrons from the gate to the channel was blocked by the low Schottky barrier formed at the PEA$_2$SnI$_4$ layer interface for Pe0 and by the high Schottky barrier formed at the PEAI+SnO$_2$ surface layer for Pe120. Overall, regardless of the polarity of $V_{GS}$, the leakage current can be suppressed by the formation of a surface layer interfacing with the gate.

Figure 4d, e shows the 2D contour maps of the resulting current density distribution in the Pe0- and Pe120-based b-JFETs, respectively, at $V_{GS}$ of −1 V (left panel) and +1 V (right panel). The arrows in the map indicate the magnitude and direction of the current density vector field. The contour maps for the current densities of Pe30, Pe60, and Pe90 are shown in Supplementary Fig. 23b. In the case of Pe0 (Fig. 4d), the vertical component of the current density corresponding to the above-mentioned current leakage pathway is substantial, particularly at the overlapping areas between the gate and drain electrodes at $V_{GS} = −1$ V or between the gate and source electrodes at $V_{GS} = +1$ V. In contrast, the vertical component of the current density for the Pe120-based device that contained the electron-blocking oxidized surface

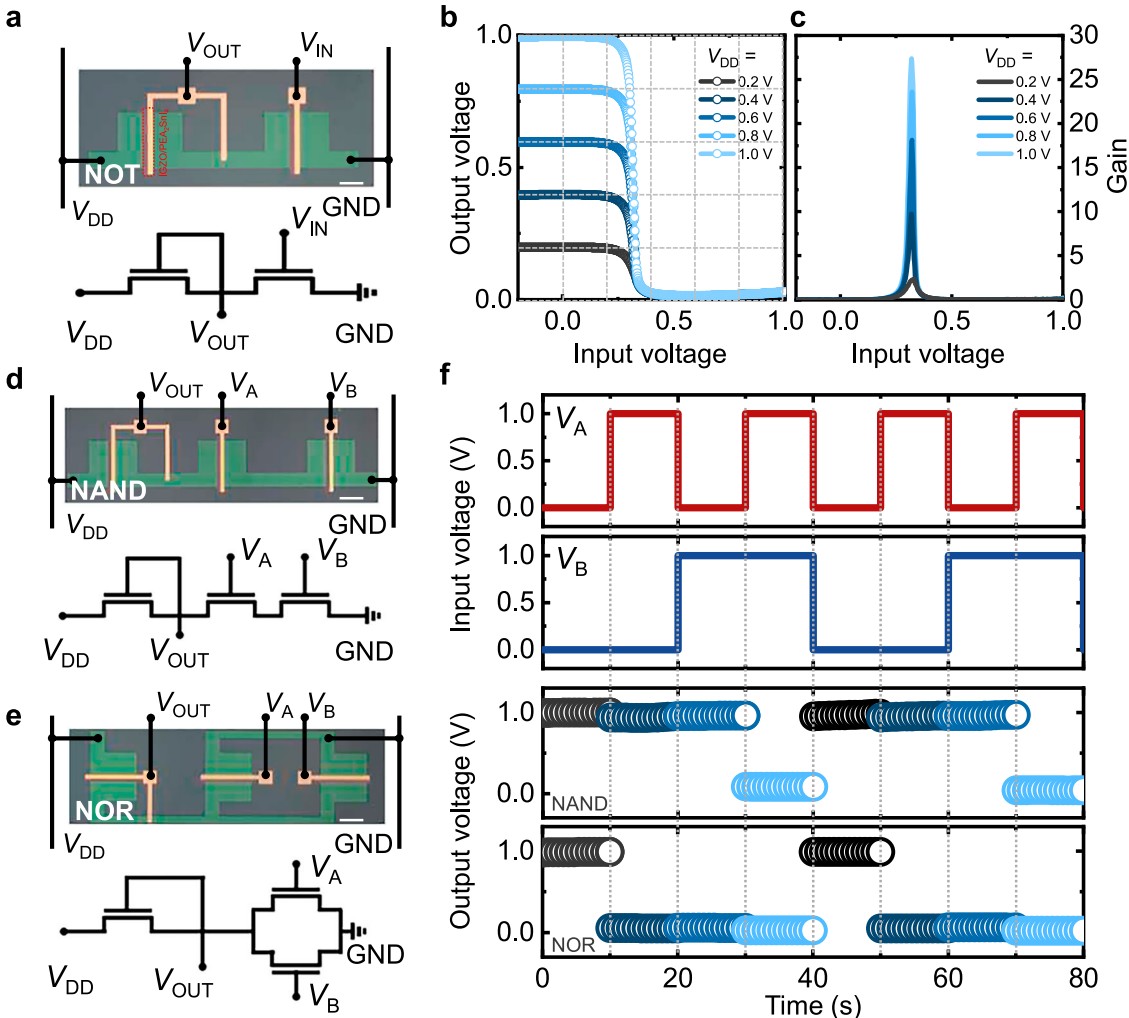

**Fig. 5 | Fabrication of logic gates. a** OM image and corresponding circuit diagram of a logic circuit for the NOT gate (scale bar = 500 μm). **b** Voltage-transfer and **c** gain characteristics of the NOT gate under various $V_{DD}$. **d**, **e** OM images and corresponding circuit diagrams of logic gates for the NAND and NOR gates at $V_{DD} = 1$ V (scale bar = 500 μm). **f** Four possible logic combinations (0,0), (1,0), (0,1), and (1,1) with the corresponding output voltages of the NAND and NOR gates.

layer is dramatically suppressed (Fig. 4e). Consequently, the lateral component dominates the current density distribution between the source and drain electrodes.

## Fabrication of logic gates

Finally, various logic gates (NOT, NAND, and NOR) were successfully fabricated using the Pe120-based b-JFET. Figure 5a shows the circuit diagram of the NOT logic gate (inverter) composed of a b-JFET connected in series with another diode-connected b-JFET serving as a pull-up device. Figure 5b shows the voltage-transfer characteristics of the NOT logic gate at various applied voltages ($V_{DD} = 0.2, 0.4, 0.6, 0.8,$ and 1.0 V). The NOT gate exhibited typical signal inversion characteristics. When the input voltage ($V_{IN}$) was in the logic state "0" ($V_{IN} = 0$ V), the output voltage ($V_{OUT}$) yielded the logic state "1" ($V_{OUT} = V_{DD}$). Conversely, when a higher $V_{IN}$ was applied, $V_{OUT}$ yielded the logic state "0." The extracted inverter gain ($dV_{OUT}/dV_{IN}$) is shown in Fig. 5c. The peak inverter gain was calculated to be 27.4 at $V_{DD} = 1$ V. In addition, the inverter exhibited low static power consumption of ~660 pW at $V_{DD} = 1$ V, owing to the enhancement-mode operation of the b-JFET that ensures negligible standby current (averaged $I_{static} \approx 6.6 \times 10^{-10}$ A when $V_{IN} = 0$ V). Figure 5d, e shows the circuit diagram of the NAND and NOR logic gates. The pull-down network of the NAND logic gate comprised two identical perovskite b-JFETs connected in series, while

that of the NOR gate comprised a b-JFET connected in series with two b-JFETs in parallel. A diode-connected b-JFET was used as a pull-up device in both logic gates. The $V_{IN}$ values of these logic gates ($V_A$ and $V_B$) were set to 0 V for the logic gate "0" and +1 V for the logic gate "1." Fig. 5f presents the $V_{OUT}$ values of the NAND and NOR gates for all four possible combinations: (0,0), (0,1), (1,0), and (1,1). The NAND gate outputs a logic "0" only when both inputs are "1"; for all other combinations, it outputs a logic "1". In contrast, the NOR gate outputs a logic "1" only when both inputs are "0"; it yields a logic "0" for any other input combination. It is worth noting that the successful fabrication of logic gates was enabled by the inherent patterability of our device, representing a key advantage over recently reported 2D flake-based JFETs[28,29,43,48,52,53], which are limited in logic implementation due to their reliance on mechanical exfoliation. This solution-processed approach, combined with controlled oxidation, provides practical opportunities for integrated circuit development and further advances in perovskite electronics.

In conclusion, we demonstrated that oxidation significantly enhanced the performance of lead-free perovskite JFETs based on a heterojunction comprising PEA$_2$SnI$_4$ and IGZO. Various chemical analyses confirmed that a simple annealing process in ambient air induced the gradual formation of sequential heterojunctions comprising SnO$_2$:PEAI, PEA$_2$SnI$_4$, and IGZO. The electronic structure evolved

distinctly as the band offsets at each heterojunction increased, causing the $I_G$ to significantly decrease by more than five orders of magnitude. This overcomes the chronic limiting factor of high leakage current in JFETs and enables unconventional JFET operations in the enhancement mode. The analysis results were successfully corroborated by computational simulations. The resulting device demonstrated high performance, achieving an average field-effect mobility of 29.4 cm²V⁻¹s⁻¹, an on/off $I_D$ ratio exceeding $10^5$, and an SS of 67.1 mV dec⁻¹. The device could achieve these values at a low operating voltage of ≤1 V by exploiting the high polarizability of the perovskite layer. Additionally, we successfully investigated the operation of NOT, NAND, and NOR logic gates using the perovskite b-JFETs. The perovskite b-JFET proposed herein provides insights into the unforeseen potential of harnessing Sn oxidation, which could open up diverse opportunities in the field of perovskite electronics.

## Methods

### Device fabrication
PEAI was purchased from Greatcell Solar. Tin(II) iodide (SnI₂; AnhydroBeads, 99.99%, metal basis), N N-dimethylformamide (DMF; anhydrous, 99.8%), and 1,3-dimethyl-3,4,5,6-tetrahydro-2(1H)-pyrimidinone (DMPU, 98%) were purchased from Merck Sigma-Aldrich. To prepare the PEA₂SnI₄ precursor, PEAI, SnI₂, and DMPU were mixed in a 2:1:1 molar ratio in DMF to achieve a concentration of 0.38 M. The resulting solution was stirred in an Ar-filled glovebox and subsequently filtered through a 0.2 μm polytetrafluorethylene filter prior to use.

A Si/SiO₂ (300 nm) wafer substrate was cleaned via successive sonication in acetone, 2-propanol, and DI water for 15 min each. A 30 nm-thick ITO layer was deposited onto the cleaned substrate via radio frequency magnetron sputtering, followed by sintering at 600 °C for 30 min in the air. The ITO layer was patterned through conventional photolithography (AZ 5214E), followed by chemical etching with 35 vol % hydrochloric acid diluted in distilled water. Next, a 10 nm IGZO layer was deposited onto the ITO layer via radio frequency magnetron sputtering, followed by sintering at 450 °C for 5 min in the air. The sintered IGZO layer was subjected to conventional photolithography using AZ 5214E and then chemically etched with 3 vol% LCE-12 (Cyantek Co.) diluted in distilled water. Thereafter, 2.5 μm-thick parylene-c was deposited via CVD and patterned via conventional lithography (AZ 5214E). Next, parylene-c was etched with reactive ion etching. The prepared PEA₂SnI₄ solution was spin-coated onto the substrate in an Ar-filled glovebox at 4000 rpm for 25 s. During the spin-coating process, 0.3 mL diethyl ether was dropped onto the spinning substrate after 8 s, and then annealing was performed at 100 °C for 10 min. The parylene-c layer was mechanically peeled off with a tweezer. Subsequently, the patterned perovskite film was thermally treated at 60 °C for 120 min in ambient air to oxidize it. Thereafter, Au (40 nm) was deposited through thermal evaporation at a rate of 1 Å s⁻¹. Finally, the device was subjected to CVD for encapsulation involving 1 μm-thick parylene-c layer deposition.

### Characterization
EIS measurements were conducted using a VersaSTAT4 potentiostat (AMETEK). XPS measurements were conducted using the VG ESCALAB 250 (Thermo Fisher Scientific) instrument equipped with a monochromatic Al-Kα radiation source ($h\nu$ = 1486.8 eV). For the depth analysis, Ar ion etching was performed using an Ar ion gun with 500 eV power and scanning intervals of 30 s. XRD measurements were conducted with high-resolution XRD (SmartLab) using Cu Kα radiation at a scan rate of 4° min⁻¹. GIXRD measurements were conducted with XRD (Bruker, D8 DISCOVER) using Cu Kα 8050 eV with the beam size of 0.1 mm. The ellipsometry was measured with the spectroscopic ellipsometry (MG-1000, Nano-view) at an incident angle of 69.8°. Hall measurements were performed using a helium cryostat with electromagnet (Closed Cycle Cryostats, Advanced Research Systems) at

300 K. The electrical properties of the transistors and logic gates were measured using a Keithley 4200A-SCS instrument. TCAD simulations were conducted using Silvaco 2D Atlas. UPS and IPES measurements were performed in an ultrahigh vacuum analysis chamber at about $10^{-10}$ Torr. The chamber was equipped with a hemispherical electron analyzer (SES-100, VG-Scienta) and a helium I discharge lamp ($\hbar\omega$ = 21.2 eV) for UPS. A low-energy electron gun and a photon detector were used for the IPES measurement. The samples were biased at −10 V for UPS to observe the secondary cutoff region for $E_{WF}$ measurements. IPES was performed in an isochromatic mode with scanning electron energy. The electron gun (PSP Vacuum Technology) emitted defined low-energy electrons onto the sample, generating light when they interacted with high-lying unoccupied states and decayed to low-lying unoccupied states. Photon detection occurred on the BaF₂ window/KBr-coated MCP photon detector (PHOTONIS). The energy scales in the UPS and IPES spectra were aligned by matching the Fermi energies ($E_F$) calibrated using a clean Au(111) crystal. The total instrumental energy resolutions were <0.1 eV for UPS and 0.5 eV for IPES.

## Data availability
The source data in this study have been deposited in the Figshare database https://doi.org/10.6084/m9.figshare.29595038. Any additional data requested will be provided by the corresponding authors.

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

## Acknowledgements

This work was supported by the Samsung Research Funding & Incubation Center of Samsung Electronics under Project Number SRFC-MA1901-52 (J.H.C).

## Author contributions

J.H. Cho and S. Oh initiated the research. J.H. Cho, S. Oh, and M.S. Kang supervised the research. S. Kim carried out most of the experimental

work and data analysis and wrote the manuscript. S.H. Kim carried out the simulation. H.U. Hwang, J. Kim, J.W. Kim, B. Kang, S. Lee, D.Y. Ryu, H. Kim, J.-M. Myoung assisted characterization. I.C. Kwak assisted with materials processing. S.B. Jo and M.S. Kang reviewed and edited the manuscript. All authors discussed the results and contributed to the writing of the manuscript.

## Competing interests

The authors declare no competing interests.
