## [Transparent Peer Review file · Nature Communications]

Revisiting the role of oxidation in stable and high-performance lead-free perovskite-IGZO junction field-effect transistors

Corresponding Author: Professor Jeong Ho Cho

Version 0:

Reviewer comments:

Reviewer #1

(Remarks to the Author)

The authors report on perovskite-IGZO junction field-effect transistors. Although some results are interesting, in my personal opinion, this manuscript does not meet the standard of Nature Communications.

1. The proposed perovskite-IGZO FET demonstrated n-type behavior with a field-effect mobility of $29.4 \text{ cm}^2\text{V}^{-1}\text{s}^{-1}$ and an on/off current ratio of >105 . However, this device performance should be easily realized by IGZO FET with much simpler process. In addition, the logic gates should also be realized by the IGZO FET.
2. The advantages of the perovskite-IGZO junction field-effect transistors have not been highlighted in the manuscript.
3. Figure 2b, how about the transfer curve at lower V_{DS} ($<0.1 \text{ V}$)?
4. P7 line 1, how about the capacitance at low frequency ($<20 \text{ Hz}$)? Please show the capacitance-frequency curve. Besides, the device mobility should be calculated at low frequency to avoid mobility overestimation.

Reviewer #2

(Remarks to the Author)

Sn-halide perovskite is an emerging low-cost and high-performance p-type semiconductor. The recent efforts on high-performance p-channel transistors demonstrate the suitability for combining with commercial n-oxides in complementary electronics applications. The authors demonstrated a PEASnI₂/IGZO JFET with good device characteristics; however, the motivation behind the study and key characterizations related to Sn oxidation and device operation remain unclear. The following questions should be addressed before this work is considered for publication in NC:

1. The introduction part should include the basic introduction of JFETs and the comparison of advantages and disadvantages with traditional MOSFETs, so that readers can understand the motivation and significance of this study. In addition, the literature on p-channel PEASnI₂ transistors is extensive and should be cited appropriately.
2. Sn-halide perovskites are prone to degradation when exposed to air. The proposed SnO₂/PEASnI₂ bilayer structure is an ideal model. I wonder if the authors could thoroughly investigate the microstructure of their air-oxidized perovskite films for the solid demonstration of bilayer structure.
3. After air oxidation, the perovskite film can degrade, leading to the formation of ion migration pathways. Please test the device under different bias conditions (e.g., varying scan speed and interval) to assess its operational reliability.
4. The PEASnI₂ thin film can fully decompose during ambient aging after several days of air exposure, changing from a brown color to transparent. I wonder the mechanism behind the high stability of JFET, which remains stable for over a month. Other solid film characterizations of perovskite film during ambient aging should be provided to understand the perovskite phase variation.
5. The practical application of JFETs using PEASnI₂ is unclear. There are several high-performance JFETs based on n-oxides and various p-type semiconductors. A benchmark comparison should be provided.

6. Traditional JFETs typically operate in depletion mode (D-JFET). In a well-designed thin-film pn heterojunction, when the bias applied to the junction is 0V, the width of the depletion layer is usually less than the thickness of the p-type and n-type thin films. For JFETs, this means that when the gate-source voltage is 0V, the device remains always-on because the conductive channel is not pinched off by the depletion region. However, the JFET discussed in this paper operates in enhancement mode (E-JFET). The authors should provide a reasonable explanation for this behavior
7. The actual energy band distribution diagram of the device cannot be accurately determined from a simple combination of UPS results for all films. The authors could provide vertical heterojunction IV curves for Pe0-IGZO and Pe120-IGZO devices and extract the Schottky barrier heights from these measurements for comparison. This would make the results more convincing
8. The carrier concentrations and mobilities of IGZO and PEASnI2 thin films measured by Hall effect should be given, which is convenient for comparison with device characteristic parameters.
9. In Fig. 5d, the author should give the time scale of the abscissa axis in order to evaluate the operating frequency of the device.

Reviewer #3

(Remarks to the Author)

The manuscript "Revisiting the role of oxidation in stable and high-performance lead-free perovskite-IGZO junction field effect transistors" by Kim et al., reported on the beneficial effect of Sn oxidation in suppressing the leakage current in the so-called barriered-JFET. Although this work was done with care, there are some fundamental issues stopping me from supporting its publication:

1. The premise of constructing a bi-layer device must be that the resultant performance is much better than the individuals, which justifies the more complex structure. However, this comparison is absent in the present study despite the vast amount of literature on both Sn-based hybrid perovskite and IGZO transistors. In fact, it is puzzling to me why and how the authors conceived the idea of making this kind of bi-layer junction device in the first place. Is it aimed at presenting the Sn oxidation? There are many ways to achieve that goal like encapsulation, designing a bilayer structure with perovskite at the bottom, and so on.
2. the thoughts on the device structure lead me to other doubts about the use of a natural oxidization layer on Sn perovskite as the dielectric layer. First, it is not surprising at all that the leakage current is high with the top gate since there is nothing there to stop it. Why not compare with the bottom gate? Second, such a natural oxidization layer will not be a good dielectric layer, much inferior to conventional ones such as SiO₂ and alumina. Why bother? In the discussion, the associated capacitance was mentioned in passing to be high, but it was not directly characterized. Why? If this thin layer is so important, it should be characterized carefully, for example using TEM to check the uniformity and crystallinity, and control experiments to check factors like oxidization temperature and duration etc.
3. in the line of scrutinizing the role of the natural oxidization layer, why not use a thin dielectric layer like PMMA or others to suppress the leakage layer? Does the perovskite layer really play any role at all? I doubt the oxidized layer brings in anything magic like ultra-high dielectric constant, super-high resistivity, or something like it.

Version 1:

Reviewer comments:

Reviewer #2

(Remarks to the Author)

The authors have addressed my concerns properly. Please see below as my comments on authors' response to Reviewer #1's concerns.

Comment 1: "The authors report on perovskite-IGZO junction field-effect transistors. Although some results are interesting, in my personal opinion, this manuscript does not meet the standard of Nature Communications.

1. The proposed perovskite-IGZO FET demonstrated n-type behavior with a field-effect mobility of 29.4 cm²V⁻¹s⁻¹ and an on/off current ratio of >10⁵. However, this device performance should be easily realized by IGZO FET with much simpler process. In addition, the logic gates should also be realized by the IGZO FET."

Opinion: The authors restate their perovskite-IGZO JFET figure of merits (low operation voltage = 1 V, SS = 67.1 mV/dec, inverter gain = 27.4, 100 °C processing) and highlight the scalable manufacturing processes. As far as I know, high-performance IGZO TFTs with low voltage operation (including mobility over 20cm²/Vs, SS value close to 60 mV/decade, and operating voltage as low as 1 V) can be realized by high-quality high-κ dielectric layers such as ALD and anodic oxidation, and the deposition temperature can also be kept low, at least similar to ~100°C proposed by this work. The scalability and manufacturability of these methods are significantly higher than that of JFETs formed by solution-processed

perovskite and IGZO proposed by the authors. In addition, the gain of pseudo-CMOS inverter composed of pure IGZO TFTs can easily exceed 50, and the performance of the inverter is not outstanding enough.

Recommendation: The authors should re-examine the real advantages of perovskite/IGZO JFETs and the scientific motivation behind this work. The authors should point out at least one or two unique advantages of their devices, which are not available in all other IGZO TFT with high performance and low voltage. In addition, the authors should provide a side-by-side table comparing their inverter (VDD, gain, SS, static power, processing temperature) with representative IGZO TFT based inverter from the literature. It is well known that IGZO technology exhibits excellent scalability, and the authors should prove that the IGZO JFETs prepared by the method proposed in this work did not lead to the loss of IGZO scalability and large-scale manufacturing. As a result, the authors should provide device yield and uniformity metrics (evaluated 100 devices on a 4-inch wafer) to justify the proposed “scalability”.

Comment 2: “The advantages of the perovskite–IGZO JFETs have not been highlighted”

Opinion: Although the introduction was rewritten to list features (low leakage, enhancement-mode, high μ /on–off, stability, logic demo), same as Comment 1, the author still has not put forward the advantages of IGZO JFET fundamentally, and the high mobility and low SS value can still be achieved by industrial compatible and high reliability ALD technology. Moreover, I do not agree with what the author claimed “sophisticated dielectric engineering”.

Recommendation: Same as Comment 1.

Comment 3: Figure 2b: transfer curve at lower VDS (< 0.1 V)?

Opinion: The authors added Supplementary Fig. 4 with different VDS values, which is good. But why the on current decrease over 3 orders of magnitude when the VDS=0.01 V (only decrease 2 orders of magnitude compared with VDS=1 V). Does this mean that the IGZO JFETs is a non-ideal device?

Recommendation: Authors should provide proper explanation of the current collapse at VDS= 0.01 V.

Comment 4: P7 line 1, how about the capacitance at low frequency (<20 Hz)? Please show the capacitance-frequency curve. Besides, the device mobility should be calculated at low frequency to avoid mobility overestimation.

Opinion: The reply discusses JFET classification and a new mobility formula, but omits crucial details on the explicit frequency range measured (< 20 Hz), the numerical values used (N_s , g_m , t), and a direct comparison of μ extracted via low-frequency capacitance versus the JFET formula.

In addition, I think it is biased to use this formula $\mu = (L/W)(g_m)/(qN_s t)$ to calculate the mobility of JFETs. This is because, N_s presented the electron concentration in the formula is obtained by Hall effect test. As we all know, the carrier concentration test result of Hall effect is influenced by many factors (such as the contact resistance of Hall electrodes, the intensity of Hall magnetic field, etc.), and the result cannot accurately reflect the absolute carrier concentration values of IGZO thin films. From the formula, when the value of N_s is doubly underrated, the mobility of the device is doubly overrated. In Hall test, it is very common that the value of N_s changes by more than one order of magnitude under different test conditions of the same sample. Thus, this method of extracting mobility should be treated with great caution.

Recommendation: The author should use three different methods to evaluate the device mobility, and compare the difference. If the difference is not so big, it can be considered that the mobility extraction is accurate. Firstly, the authors should recalculate the mobility using conventional C–V method at different frequencies. Secondly, the four-probe Hall test could be used to obtain the Hall mobility of IGZO thin films with the same thickness in JFETs. Thirdly, authors should use the same IGZO process and film thickness as JFET in this work to prepare IGZO MOSFETs on reliable SiO₂ dielectric, and use the calculation method of MOSFETs to obtain the device field effect mobility. By comparing these methods, the mobility extraction can be more credible.

Reviewer #3

(Remarks to the Author)

The revised work demonstrates a commendable effort to refine and enhance the original draft. The improvements in clarity and depth of analysis are evident, particularly in comparison of other literature on Sn-based perovskite and IGZO transistors and clarification of the function of a natural oxidation layer on Sn perovskites. In general, this revision marks significant progress toward achieving the work's objectives.

Version 2:

Reviewer comments:

Reviewer #2

(Remarks to the Author)

I have carefully reviewed the responses and the revised manuscript. The authors have made considerable efforts to address the key scientific and technical questions that were previously raised. Specifically, they have clarified the claimed advantages of the perovskite-IGZO JFETs. The authors have also improved the quantitative support for their device scalability. For the technical concerns regarding the current suppression at low VDS and the use of Hall-derived carrier concentration in the mobility formula, the authors have provided reasonable justifications and supplemental data.

The manuscript now meets the standard of Nature Communications. I recommend acceptance.

Reviewer #3

(Remarks to the Author)

The authors have shown lots of effort in revising the manuscript, which is acceptable, I think.

REVIEWER COMMENTS

Reviewer #1

The authors report on perovskite-IGZO junction field-effect transistors. Although some results are interesting, in my personal opinion, this manuscript does not meet the standard of Nature Communications.

1. The proposed perovskite-IGZO FET demonstrated n-type behavior with a field-effect mobility of 29.4 cm²V⁻¹s⁻¹ and an on/off current ratio of >10⁵. However, this device performance should be easily realized by IGZO FET with much simpler process. In addition, the logic gates should also be realized by the IGZO FET.

Reply: We appreciate the comment regarding our perovskite-IGZO JFET devices. While we acknowledge that our device performance metrics - including the field-effect mobility of 29.4 cm²/V·s and on/off current ratio exceeding 10⁵ - are indeed comparable to conventional IGZO FETs with simpler structures, we would like to highlight several distinctive advantages of our approach.

Our device achieves a remarkably low subthreshold swing of 67.1 mV dec⁻¹ and operates within a narrow voltage range of -1 V to 1 V, making it particularly well-suited for low-power applications. We recognize that some high-performance IGZO FETs using high-*k* dielectric layers have demonstrated comparable performance metrics. However, these devices typically rely on sophisticated fabrication methods like atomic layer deposition (ALD), which involve complex processes and high-temperature requirements (**Revision Table 2**). In contrast, our perovskite-IGZO JFET employs a straightforward solution-processing approach, utilizing simple spin-coating for the perovskite layer followed by mild thermal annealing at just 100°C. This method significantly enhances manufacturability and scalability. Additionally, our device demonstrates excellent patterning compatibility, enabling the successful implementation of logic gates. The combination of low operating voltage and small subthreshold swing has allowed us to achieve an inverter gain of 27.4, as detailed in our manuscript. This performance notably surpasses typical IGZO-based logic circuits in terms of both efficiency and power consumption.

These distinct advantages position our device as a promising candidate for applications demanding low-power operation, high performance, and scalable manufacturing processes. We believe these aspects effectively address the concerns of reviewers while emphasizing the unique contributions of our work. We have completely changed the introduction and embedded these points in the introduction with references from **Revision Table 2** to elaborate our point.

Introduction:

Organometallic halide perovskites have gained significant attention owing to their tunable bandgap, facile processability, and high carrier mobility¹⁻³. These properties have enabled major advances in optoelectronics, particularly in solar cells and light-emitting diodes, while also opening new possibilities for transistor applications⁴⁻¹¹. Although Pb-based perovskites have demonstrated the most excellent optoelectronic properties, their toxicity poses serious environmental concerns and limits industrial applications¹²⁻¹⁴. Sn-based perovskites have emerged as promising alternatives due to their similar valence electron configuration^{11,15-17}. However, susceptibility of Sn to oxidation due to the absence of the lanthanide shrinkage effect, transitioning from Sn²⁺ to Sn⁴⁺, has hindered their practical implementation¹⁸⁻²⁰. Previous research has shown that Sn-based perovskites demonstrate highly promising performance as the *p*-type channel material in thin-film transistors; however, even the slightest exposure to air within mere minutes leads to self-*p*-doping, which results in metallic characteristics that hinder their function as transistors^{4,11,21-25}. While numerous research efforts have focused on suppressing this oxidation through various chemical and physical strategies, these approaches face inherent limitations^{26,27}. Considering the spontaneity of Sn oxidation and distinctive evolution in its electronic structures, the unique electronic properties of oxidized Sn-based perovskites could potentially create new opportunities in electronic devices.

The increasing prevalence of Internet of Things (IoT) devices has made low power consumption a critical requirement in modern electronics. Junction field-effect transistors (JFETs) have emerged as a promising candidate for such applications due to their steep subthreshold swing values^{28,29}. Unlike conventional metal-oxide-semiconductor field-effect transistors (MOSFETs), which require sophisticated dielectric engineering to approach the theoretical limit of 60 mV dec⁻¹ at 300K³⁰⁻⁴², JFETs can theoretically achieve this performance more readily due to their inherently high gate capacitance achieved without a gate dielectric layer²⁹. However, this architecture presents two significant challenges: substantial gate leakage

current and operation limited to depletion mode. The depletion-mode constraint is particularly problematic for power consumption since these devices conduct significant current at zero gate voltage. Furthermore, achieving enhancement mode operation in conventional JFETs is fundamentally limited by large forward bias currents at the PN junction^{43,44}. Therefore, developing JFETs capable of enhancement mode operation while maintaining low leakage current remains a crucial challenge for advancing low-power electronics.

In this study, we successfully fabricated a stable and high-performance transistor by tailoring and exploiting the oxidation phenomenon of Sn-based perovskites. Specifically, a JFET was fabricated with *p*-type PEA₂SnI₄ and *n*-type indium gallium zinc oxide (IGZO), which formed a PN junction. While the as-prepared JFET exhibited poor performance due to significant gate leakage current, the JFET based on the perovskite layer with an oxidized surface exhibited dramatic changes in its operation. By exploiting the oxidized layer instead of preventing its formation, the gate leakage current of the JFET was suppressed below 10⁻¹⁰ A. Moreover, the JFET was able to operate unconventionally in the enhancement mode owing to the suppression of gate leakage current by the oxidation layer, thereby overcoming one of the fundamental limiting factors of conventional JFETs. We refer to such a device as a barriered JFET (b-JFET). Owing to the high polarizability of its perovskite layer, the resulting perovskite b-JFET achieved an averaged field-effect mobility of 29.4 cm²V⁻¹s⁻¹, a low subthreshold swing (SS) of 67.1 mV dec⁻¹, and a high on/off current ratio of >10⁵ at a voltage of ≤1 V. Moreover, the device exhibited excellent bias stress stability, operational stability, and environmental stability over a month in ambient air. Additionally, we successfully constructed logic circuits and achieved high inverter gain values, further confirming the potential of the device for various practical applications.

Revision Table 1. Performance comparison between the perovskite-employed field effect transistors.

Employed as	Composition	Dielectric	Mobility (cm ² V ⁻¹ s ⁻¹)	Operation voltage (V)	Air stability	Ref
Channel	FACsPEASnI	SiO ₂	17.4	-40 to 40	-	[21]
Channel	CsFAPEASnI	SiO ₂	70	-40 to 40	2 min*	[22]
Channel	PEAFASnI	PMMA-Al ₃ O ₃	0.21	-15 to 5	-	[11]
Channel	CsSnI ₃	SiO ₂	50	-40 to 40	1 min*, 36 hours	[23]
Channel	FPEASnI	SiO ₂	2.96	-40 to 10	30 min	[4]
Channel	FAPEASnI	HfO ₂	12	-8 to 8	-	[24]
Channel	(BA) ₂ SnI ₄	PVA/CL-PVP	0.03	-60 to 20	-	[25]
Gate	PEA₂SnI₄	-	29.4	-1 to 1	1 month	This work

*unencapsulated

Revision Table 2. Performance comparison between the solution processed perovskite-IGZO JFET and previously reported IGZO-based transistors with high-k dielectrics

Type	Dielectric (perovskite) deposition method	Channel material	Subthreshold swing (mV dec ⁻¹)	Process temperature (°C)	Mobility (cm ² V ⁻¹ s ⁻¹)	On/off ratio	Operating voltage (V)	Ref
MOSFET	ALD	IGZO	62.29	400	18.9	2.8 × 10 ⁹	-1 to 3	[30]
	ALD	IGZO	70.2	250	55.3	~ 10 ⁹	-1 to 3	[31]
	PEALD	IGZO	76	350	6.1	3.5 × 10 ⁹	-3 to 10	[32]
	ALD	IGZO	87	300	19.7	4.3 × 10 ¹¹	-1 to 4	[33]
	RF sputtering	IGZO	88	150	2.3	5 × 10 ⁵	0 to 1	[34]
	ALD	IGZO	90	150	10.4	2.8 × 10 ⁹	-5 to 5	[35]
	PEALD	IGZO	130	250	21.7	3.2 × 10 ⁸	-2 to 2	[36]
	Solution spin-coating	IGZO	140	350	85	~ 10 ⁴	-2 to 2	[37]
	ALD	IGZO	170	150	15.1	-	-5 to 5	[38]
	E-beam	IGZO	250	RT	61.5	~ 10 ⁵	-2 to 3	[39]

	ALD	IGZO	256	150	9.7	1.3×10^6	-2 to 10	[40]
	Printing	IGZO	300	150	4.3	10^8	-4 to 4	[41]
	RF sputtering	IGZO	560	RT	28	10^7	-6 to 10	[42]
JFET	Solution spin-coating	IGZO	67.1	100	29.4	1.2×10^5	-1 to 1	This work

2. The advantages of the perovskite-IGZO junction field-effect transistors have not been highlighted in the manuscript.

Reply: We appreciate the comment. We completely rewrote the introduction of the manuscript to include the advantages of the perovskite- bJFETs. As shown in the new introduction, these advantages include:

1. Low SS (67.1 mV dec^{-1}) is easily achieved without sophisticated dielectric engineering.
2. By exploiting the oxidized layer instead of preventing its formation, the gate leakage current of the JFET was suppressed below 10^{-10} A
3. The JFET was able to operate unconventionally in the accumulation mode owing to the suppression of gate leakage current by the oxidation layer, thereby overcoming one of the fundamental limiting factors of conventional JFETs
4. High performance metrics: averaged field-effect mobility of $29.4 \text{ cm}^2\text{V}^{-1}\text{s}^{-1}$, high on/off current ratio of $>10^5$ at a low voltage of $\leq 1 \text{ V}$
5. Excellent stability characteristics: good bias stress stability, operational stability, and environmental stability over a month in ambient air
6. Successfully demonstrated practical applications through logic circuit implementation with high inverter gain values

These advantages comprehensively address both the fundamental limitations of conventional JFETs and the practical requirements for modern electronic devices, particularly in low-power applications.

3. Figure 2b, how about the transfer curve at lower V_{DS} ($<0.1 \text{ V}$)?

Reply: We appreciate the comment regarding the device characteristics at lower drain voltages. To address this comment, we conducted additional measurements and included **Supplementary Fig. 4** which shows the transfer curves under various drain voltages ($V_{DS} = 1, 0.5, 0.1, 0.05, \text{ and } 0.01 \text{ V}$).

Page 6, line 23: The resulting device characteristics at low drain voltages showed comparable performance to those at high drain voltages with an on/off-state current ratio above 10^5 and an electron mobility (μ) of $29.4 \text{ cm}^2\text{V}^{-1}\text{s}^{-1}$ at low voltages of $\leq 1 \text{ V}$ (transfer characteristics measured at different drain voltages are presented in **Supplementary Fig. 4**).

Supplementary Fig. 4. Transfer characteristics of the perovskite-bJFET measured at different drain voltages ($V_{DS} = 1, 0.5, 0.1, 0.05 \text{ and } 0.01 \text{ V}$).

4. P7 line 1, how about the capacitance at low frequency (<20 Hz)? Please show the capacitance-frequency curve. Besides, the device mobility should be calculated at low frequency to avoid mobility overestimation.

Reply: We appreciate the comments regarding the capacitance measurements and mobility calculations. To address these points, we have conducted the capacitance (C)-frequency (f) measurements in **Supplementary Figure 6**, which shows the C - f characteristics for both Au-perovskite-ITO and Au-perovskite-IGZO-ITO structures.

The Bode plot (**Supplementary Fig. 6a-c**) shows that the phase angle converges to 0 degree for the device without IGZO. This indicates that neither the perovskite layer nor the perovskite-oxidation layer exhibits dielectric behavior; instead, it functions merely as a resistor. In contrast, the device incorporating IGZO (**Supplementary Fig. 6d-f**) exhibits distinctly different behavior, characterized by a non-zero phase angle. This observation aligns with the semicircle observed in the Nyquist plot (**Supplementary Fig. 17**). The semicircle observed at intermediate frequency (10 - 10^4 Hz) region can be attributed to charge accumulation at the perovskite-IGZO interface¹. These electrical characteristics are fundamental to the operation of the device and support our classification of the device as a JFET rather than a MOSFET. Consequently, we applied a JFET mobility calculation method, which is independent of capacitance measurement, avoiding potential mobility overestimation.

For accurate mobility determination of a JFET, we employed a direct calculation method based on the equation $\mu = (L/W)(g_m)/(q \cdot N_s \cdot t)$, where L/W represents the channel length-to-width ratio, g_m is the maximum transconductance, q denotes the element charge, N_s is the electron density determined through separate Hall measurements, and t is the channel thickness. The detailed discussion of mobility estimation was presented in the characterization section. However, to enhance clarity and avoid misunderstandings, we have relocated the mobility characterization discussion from the characterization section to the main text, and we also have incorporated a discussion of **Supplementary Fig. 6**, providing analysis of capacitance-frequency characteristics and their implications for device operation.

Page 6, line 26: The field-effect mobility (μ) of our devices was calculated using the equation $\mu = (L/W)(g_m)/(q \cdot N_s \cdot t)$, where L/W , g_m , q , N_s , and t denote the channel length-to-width ratio, maximum transconductance defined as $\partial I_D/\partial V_G$, element charge, electron density estimated from a separate Hall measurement (**Supplementary Fig. 5**), and channel thickness, respectively. Note that the mobility calculation does not rely on the capacitance measurements, as we employ the JFET mobility calculation equation rather than the conventional MOSFET equation. This classification of our device as a JFET is supported by our impedance analysis (**Supplementary Fig. 6**), which shows that neither the perovskite layer nor the perovskite-oxide layer functions as a conventional dielectric.

Supplementary Fig. 6. Frequency-dependent capacitance and phase angle measurements of perovskite devices. (a) Schematic of Au-perovskite-ITO device structure. (b), (c) Bode plots showing the capacitance and phase angle *versus* frequency for Pe0 and Pe120, respectively. (d) Schematic of Au-perovskite-IGZO-ITO

device structure (e-f) Bode plots showing the capacitance and phase angle *versus* frequency for Pe0 and Pe120, respectively.

Analysis of the Bode plots demonstrates fundamental differences in device behavior with and without IGZO integration. The phase angle convergence to 0 degree in devices without IGZO (**Supplementary Fig. 6a-c**) indicates a purely resistive characteristic. Conversely, IGZO-incorporated devices (**Supplementary Fig. 6d-f**) exhibit capacitive behavior with a non-zero phase angle, correlating with the semicircular response in the Nyquist plot (**Supplementary Fig. 17**). The observed semicircle in the intermediate frequency range (10 - 10^4 Hz) indicates charge accumulation at the perovskite-IGZO interface¹. Based on these electrical characteristics, we classified our device as a JFET and accordingly employed a capacitance-independent mobility calculation method to ensure accurate mobility assessment.

References

1. Abdulrahim, S. M., Ahmad, Z., Bahadra, J. & Al-Thani, N. J. Electrochemical impedance spectroscopy analysis of hole transporting material free mesoporous and planar perovskite solar cells. *Nanomaterials* **10**, 1635 (2020).

Reviewer #2

Sn-halide perovskite is an emerging low-cost and high-performance p-type semiconductor. The recent efforts on high-performance p-channel transistors demonstrate the suitability for combining with commercial n-oxides in complementary electronics applications. The authors demonstrated a PEASnI₂/IGZO JFET with good device characteristics; however, the motivation behind the study and key characterizations related to Sn oxidation and device operation remain unclear. The following questions should be addressed before this work is considered for publication in NC:

1. The introduction part should include the basic introduction of JFETs and the comparison of advantages and disadvantages with traditional MOSFETs, so that readers can understand the motivation and significance of this study. In addition, the literature on p-channel PEASnI₂ transistors is extensive and should be cited appropriately.

Reply: We appreciate the valuable feedback regarding the manuscript's introduction. While our original introduction emphasized our innovative approach of transforming a common limitation of Sn perovskite (susceptibility to oxidation) into an advantage for electronic devices, we acknowledge that we did not provide sufficient background information about JFETs. To address this, we have thoroughly revised the introduction to include a comprehensive discussion of JFET fundamentals and their advantages over MOSFET, incorporating references presented from **Revision Table 2**, which compares our device performance with IGZO-based MOSFETs. Additionally, following the reviewer's suggestion, we expanded our discussion of perovskite-based transistors, adding references from **Revision Table 1** that compares device performance metrics with other existing perovskite-based transistors which employed perovskites as a *p*-type channel.

Introduction:

Organometallic halide perovskites have gained significant attention owing to their tunable bandgap, facile processability, and high carrier mobility¹⁻³. These properties have enabled major advances in optoelectronics, particularly in solar cells and light-emitting diodes, while also opening new possibilities for transistor applications⁴⁻¹¹. Although Pb-based perovskites have demonstrated the most excellent optoelectronic properties, their toxicity poses serious environmental concerns and limits industrial applications¹²⁻¹⁴. Sn-based perovskites have emerged as promising alternatives due to their similar valence electron configuration^{11,15-17}. However, susceptibility of Sn to oxidation due to the absence of the lanthanide shrinkage effect, transitioning from Sn²⁺ to Sn⁴⁺, has hindered their practical implementation¹⁸⁻²⁰. Previous research has shown that Sn-based perovskites demonstrate highly promising performance as the *p*-type channel material in thin-film transistors; however, even the slightest exposure to air within mere minutes leads to self-*p*-doping, which results in metallic characteristics that hinder their function as transistors^{4,11,21-25}. While numerous research efforts have focused on suppressing this oxidation through various chemical and physical strategies, these approaches face inherent limitations^{26,27}. Considering the spontaneity of Sn oxidation and distinctive evolution in its electronic structures, the unique electronic properties of oxidized Sn-based perovskites could potentially create new opportunities in electronic devices.

The increasing prevalence of Internet of Things (IoT) devices has made low power consumption a critical requirement in modern electronics. Junction field-effect transistors (JFETs) have emerged as a promising candidate for such applications due to their steep subthreshold swing values^{28,29}. Unlike conventional metal-oxide-semiconductor field-effect transistors (MOSFETs), which require sophisticated dielectric engineering to approach the theoretical limit of 60 mV dec⁻¹ at 300K³⁰⁻⁴², JFETs can theoretically achieve this performance more readily due to their inherently high gate capacitance achieved without a gate dielectric layer²⁹. However, this architecture presents two significant challenges: substantial gate leakage current and operation limited to depletion mode. The depletion-mode constraint is particularly problematic for power consumption since these devices conduct significant current at zero gate voltage. Furthermore, achieving enhancement mode operation in conventional JFETs is fundamentally limited by large forward bias currents at the PN junction^{43,44}. Therefore, developing JFETs capable of enhancement mode operation while maintaining low leakage current remains a crucial challenge for advancing low-power electronics.

In this study, we successfully fabricated a stable and high-performance transistor by tailoring and exploiting the oxidation phenomenon of Sn-based perovskites. Specifically, a JFET was fabricated with *p*-type PEA₂SnI₄ and *n*-type indium gallium zinc oxide (IGZO), which formed a PN junction. While the as-prepared JFET exhibited poor performance due to significant gate leakage current, the JFET based on the

perovskite layer with an oxidized surface exhibited dramatic changes in its operation. By exploiting the oxidized layer instead of preventing its formation, the gate leakage current of the JFET was suppressed below 10^{-10} A. Moreover, the JFET was able to operate unconventionally in the enhancement mode owing to the suppression of gate leakage current by the oxidation layer, thereby overcoming one of the fundamental limiting factors of conventional JFETs. We refer to such a device as a barriered JFET (b-JFET). Owing to the high polarizability of its perovskite layer, the resulting perovskite b-JFET achieved an averaged field-effect mobility of $29.4 \text{ cm}^2 \text{ V}^{-1} \text{ s}^{-1}$, a low subthreshold swing (SS) of 67.1 mV dec^{-1} , and a high on/off current ratio of $>10^5$ at a voltage of ≤ 1 V. Moreover, the device exhibited excellent bias stress stability, operational stability, and environmental stability over a month in ambient air. Additionally, we successfully constructed logic circuits and achieved high inverter gain values, further confirming the potential of the device for various practical applications.

Revision Table 1. Performance comparison between the perovskite-employed field effect transistors.

Employed as	Composition	Dielectric	Mobility ($\text{cm}^2 \text{ V}^{-1} \text{ s}^{-1}$)	Operation voltage (V)	Air stability	Ref
Channel	FACsPEASnI	SiO ₂	17.4	-40 to 40	-	[21]
Channel	CsFAPEASnI	SiO ₂	70	-40 to 40	2 min*	[22]
Channel	PEAFASnI	PMMA-Al ₂ O ₃	0.21	-15 to 5	-	[11]
Channel	CsSnI ₃	SiO ₂	50	-40 to 40	1 min*, 36 hours	[23]
Channel	FPEASnI	SiO ₂	2.96	-40 to 10	30 min	[4]
Channel	FAPEASnI	HfO ₂	12	-8 to 8	-	[24]
Channel	(BA) ₂ SnI ₄	PVA/CL-PVP	0.03	-60 to 20	-	[25]
Gate	PEA₂SnI₄	-	29.4	-1 to 1	1 month	This work

* unencapsulated

Revision Table 2. Performance comparison between the solution processed perovskite-IGZO JFET and previously reported IGZO-based transistors with high-k dielectrics.

Type	Dielectric (perovskite) deposition method	Channel material	Subthreshold swing (mV dec ⁻¹)	Process temperature (°C)	Mobility ($\text{cm}^2 \text{ V}^{-1} \text{ s}^{-1}$)	On/off ratio	Operating voltage (V)	Ref
MOSFET	ALD	IGZO	62.29	400	18.9	2.8×10^9	-1 to 3	[30]
	ALD	IGZO	70.2	250	55.3	$\sim 10^9$	-1 to 3	[31]
	PEALD	IGZO	76	350	6.1	3.5×10^9	-3 to 10	[32]
	ALD	IGZO	87	300	19.7	4.3×10^{11}	-1 to 4	[33]
	RF sputtering	IGZO	88	150	2.3	5×10^5	0 to 1	[34]
	ALD	IGZO	90	150	10.4	2.8×10^9	-5 to 5	[35]
	PEALD	IGZO	130	250	21.7	3.2×10^8	-2 to 2	[36]
	Solution spin-coating	IGZO	140	350	85	$\sim 10^4$	-2 to 2	[37]
	ALD	IGZO	170	150	15.1	-	-5 to 5	[38]
	E-beam	IGZO	250	RT	61.5	$\sim 10^5$	-2 to 3	[39]
	ALD	IGZO	256	150	9.7	1.3×10^6	-2 to 10	[40]
	Printing	IGZO	300	150	4.3	10^8	-4 to 4	[41]
	RF sputtering	IGZO	560	RT	28	10^7	-6 to 10	[42]
JFET	Solution spin-coating	IGZO	67.1	100	29.4	1.2×10^5	-1 to 1	This work

2. Sn-halide perovskites are prone to degradation when exposed to air. The proposed SnO₂/PEASnI₂ bilayer structure is an ideal model. I wonder if the authors could thoroughly investigate the microstructure of their

air-oxidized perovskite films for the solid demonstration of bilayer structure.

Reply: We appreciate the suggestion regarding the thorough investigation of the oxidized perovskite film microstructure. To address this important point, we additionally carried out XPS depth profile analysis depending on annealing time, ellipsometry analysis, and grazing incidence X-ray diffraction (GIXRD). Furthermore, we attempted cross-sectional transmission electron microscopy (TEM) analysis, although direct observation was not successful. Details of these analyses are provided below.

First, we have expanded our XPS depth profile analysis beyond original Pe0 and Pe120 samples to include Pe60 and Pe90 films, allowing a more comprehensive temporal resolution of the oxidation process. Through analysis of O₂ peak areas and Sn⁴⁺/Sn²⁺ deconvoluted peak area ratios at various depths, we have verified the progressive oxidation where the oxidation layer begins to form at the surface and gradually develops over time. We have added the following sentences and **Supplementary Fig. 13** in the manuscript:

Page 8, line 26: To understand the influence of thermal treatment of PEA₂SnI₄ layer, we examined the chemical composition of the surface layers of Pe0, Pe60, Pe90 and Pe120 using X-ray photoelectron spectroscopy (XPS) depth profile analysis upon etching with 500 eV monoatomic Ar ions (**Figure 3a** and **Supplementary Fig. 13** for Pe60 and Pe90).

Supplementary Fig. 13 (a) XPS depth profiles of Pe0, Pe60, Pe90, and Pe120 films. (b) Heat map showing oxygen area of Pe0, Pe60, Pe90, and Pe120 as a function of the etch depth. (c) Depth-dependent Sn⁴⁺-to-Sn²⁺ ratios for Pe0, Pe60, Pe90 and Pe120 calculated from the deconvoluted peaks of the Sn 3d XPS spectra.

Second, the progression of the bilayer structure was further verified through ellipsometry measurements, which confirmed the formation and growth of the surface oxidation layer. Using the Forouhi-Bloomer model to fit the thickness of both the bulk perovskite and surface layer, we obtained distinct differences in oxidation layer thickness - oxidation layer / bulk perovskite layer thicknesses 0 nm / 161.3 nm (Pe0), 8.5 nm / 155.1 nm (Pe60), 11.7 nm / 138.3 nm (Pe90), and 16.8 nm / 130.5 nm (Pe120) - that corroborates well with our other analyses. We have included these results in **Supplementary Table 1** and **Supplementary Fig. 14** with additional details as follows:

Page 9, line 13: Moreover, ellipsometry measurements of Pe0, Pe60, Pe90, and Pe120 films were conducted to further elucidate the progression of the bilayer structure, confirming the formation and growth of the surface layer as annealing time increased (**Supplementary Table 1** and **Supplementary Fig. 14**).

Supplementary Fig. 14. Ellipsometry measurements showing angles of ψ and δ of Pe0, Pe60, Pe90 and Pe120 as a function of wavelength in a range of 550-820 nm.

The ellipsometry data was fitted using the Forouhi-Bloomer model to determine the thickness of both the bulk perovskite and surface oxidation layers^{2,3}. The analysis employed the following equations:

$$n(E) = n_{\infty} + \sum_{k=1}^N \frac{B_0 k E + C_0 k}{E^2 - B_k E + C_k}, \dots \dots \dots \text{(eq.1)}$$

$$k(E) = \sum_{k=1}^N \frac{A_k (E - E_g)^2}{E^2 - B_k E + C_k}, \dots \dots \dots \text{(eq.2)}$$

where n and k are real and imaginary components of the refractive index, respectively, both being functions of photon energy E . E_g is the bandgap, A_i , B_i , C_i are constants related to i) the square of the position matrix element (electron transition lifetime), ii) twice the bandwidth difference between the conduction band and the valence band, iii) dependent variable based on A_i and B_i , respectively. The fitting parameters for PEA₂SnI₄ films are detailed in **Supplementary Table 1**.

The analysis revealed progressive growth of the oxidation layer: 0 nm (Pe0), 8.5 nm (Pe60), 11.7 nm (Pe90), and 16.8 nm (Pe120). In contrast, the bulk PEA₂SnI₄ layer thickness showed a decreasing trend: 163.1 nm (Pe0), 155.1 nm (Pe60), 138.3 nm (Pe90), and 130.5 nm (Pe120).

References

- 2 S. Cai *et al.*, Fast-Response Oxygen Optical Fiber Sensor based on PEA₂SnI₄ Perovskite with Extremely Low Limit of Detection. *Adv. Sci.*, **9**, 2104708 (2022)
- 3 Forouhi A., Bloomer I., Optical dispersion relations for amorphous semiconductors and amorphous dielectrics. *Phys. Rev. B*, **34**, 7018 (1986)

Supplementary Table 1. Fitting parameters for ellipsometry measurements.

Model parameter	PEA ₂ SnI ₄
n_{∞}	2.069
E_g	1.934
A_1	0.172
B_1	4.044
C_1	4.090
A_2	0.135
B_2	10.624
C_2	27.340

To further elucidate the microstructural evolution of the surface layer, we conducted grazing incidence X-ray diffraction (GIXRD) measurements. At a grazing incidence angle (α) of 0.7° , we observed a new peak at $2\theta = 12.88^\circ$, which was not observed in the original XRD results (**Supplementary Fig. 15**). The intensity of this peak exhibited a progressive increase from Pe0 to Pe120. Based on the references 4-6, we attribute this characteristic peak to SnI_2 , which is an intermediate product during the oxidation process of PEA_2SnI_4 . The SnI_2 peak was undetectable in the conventional XRD measurement perhaps due to its relatively low crystallinity compared to the dominant signal from the highly crystalline bulk PEA_2SnI_4 . To further validate our findings, we conducted an analysis of peak intensity at $2\theta = 12.88^\circ$ across incident angles ranging from 0° to 1° . Substantial peak intensity was monitored at higher α values for films oxidized for longer period of time, indicating that the surface layer yielding SnI_2 diffraction progressively grows with increased oxidation time. . We have added the following discussion and **Supplementary Fig. 9** to elaborate on these findings:

Page 9, line 18: Additionally, grazing incidence X-ray diffraction measurements were conducted on these films (**Supplementary Fig. 16**), also revealing the progressive growth of the surface layer.

Supplementary Fig. 16. GIXRD results of Pe0, Pe60, Pe90 and Pe120 films. (a) GIXRD data showing the evolution of the peak at $2\theta = 12.88^\circ$ across Pe0, Pe60, Pe90, Pe120 samples, measured at a grazing incidence angle (α) of 0.7° . (b) Peak intensity at $2\theta = 12.88^\circ$ as a function of incident angle α .

The GIXRD measurements (**Supplementary Fig. 16a**) revealed a distinctive peak at $2\theta = 12.88^\circ$ when using a grazing incidence angle (α) of 0.7° , which was not detected in conventional XRD measurements perhaps due to the relatively low crystallinity of the component compared to the highly crystalline bulk PEA_2SnI_4 (**Supplementary Fig. 15**). The intensity of the peak showed a progressive increase from Pe0 to Pe120. We attribute this unobserved peak to $\text{SnI}_2^{4,5}$, which is presumably an intermediate product in the sequential oxidation reaction pathway of $\text{PEA}_2\text{SnI}_4^6$. Furthermore, we conducted an analysis of peak intensity at $2\theta = 12.88^\circ$ across incident angles ranging from 0° to 1° (**Supplementary Fig. 16b**). Substantial peak intensity was monitored at higher α values for films oxidized for longer period, indicating that the surface layer yielding SnI_2 diffraction progressively grows with increased oxidation time. These results altogether provide strong evidence for the progressive growth of the surface layer with increasing oxidation time.

References

- 4 Chen, Y. -S. *et al.*, Intermediate-Controlled Synthesis of Quasi-2D $(\text{PEA})_2\text{MA}_4\text{Pb}_5\text{I}_{16}$ in the 20–30% Relative Humidity Glovebox Environment for Fabricating Perovskite Solar Cells with 1 Month Durability in the Air. *ACS Omega*, **9**, 48374-48389 (2024)
- 5 Sembito, A. *et al.*, Characterization of 2D- PEA_2SnI_4 Perovskite Thin Films Grown by Sequential Physical Vapor Deposition. *Vacuum*, **233**, 113954 (2025)
- 6 Ju, Y. *et al.*, The Evolution of Photoluminescence Properties of PEA_2SnI_4 Upon Oxygen Exposure: Insight into Concentration Effects. *Adv. Funct. Mater.*, **32**, 2108296 (2021)

Furthermore, we attempted cross-sectional TEM analysis of the oxidized perovskite films to directly examine their microstructure and confirm the bilayer formation. Because perovskite films are prone to oxidate, we had to find a setup where sample treated with a focus-ion-beam (FIB) can be directly transferred to a TEM without exposure to ambient condition. Despite the use of such setup, we were unable to clearly observe the bilayer structure, likely due to several factors. Of all, it was found that the high-energy ion beam used to form the cross-section sample caused structural damage such as amorphization or ion implantation, disrupting the crystal structure of perovskites. In fact, lattice fringe patterns of the lattice were not observed even from the bulk PEA_2SnI_4 perovskite film. Therefore, carrying out structural analysis of the oxidized surface layer was not possible. Moreover, the Pt coating applied to protect the perovskite surface may have diffused or become embedded into the surface layer of the film, obscuring the thin bilayer structure and making it difficult to visualize it from TEM imaging. We show EDS elemental mapping.

Revision Fig. 1. TEM and EDS elemental mapping.

3. After air oxidation, the perovskite film can degrade, leading to the formation of ion migration pathways. Please test the device under different bias conditions (e.g., varying scan speed and interval) to assess its operational reliability.

Reply: We appreciate the suggestion regarding the investigation of operational reliability under various bias conditions. We have investigated the effect of scan speed and intervals on the transfer curve shape (**Revision Fig. 1**). Across the range of measurement conditions accessible with our setup, we observed remarkable consistency in the transfer characteristics, with no significant variations in curve shape. This stability in electrical response demonstrates the excellent operational reliability of our device.

The reviewer raises a valid point, as halide ion migration is widely known as a fundamental degradation mechanism in the perovskite community. However, the excellent operational stability of our device, even after oxidation, provides strong evidence that ion migration does not substantially influence the device operation mechanism.

Revision Fig. 2. Effect of measurement speed and interval on device transfer characteristics.

4. The PEASnI_2 thin film can fully decompose during ambient aging after several days of air exposure, changing from a brown color to transparent. I wonder the mechanism behind the high stability of JFET, which remains stable for over a month. Other solid film characterizations of perovskite film during ambient aging should be provided to understand the perovskite phase variation.

Reply: We appreciate the valuable feedback regarding the stability of PEA_2SnI_4 . Indeed, untreated PEA_2SnI_4 films undergo complete degradation after several days of air exposure, ultimately transitioning from brown color to transparent as the reviewer noted.

However, our UV-vis spectroscopy analysis (**Supplementary Fig. 9**) reveals a crucial finding: the rate of absorbance change (which is an indicative of PEA_2SnI_4 degradation) reduces over time with oxidation duration. This observation suggests that the SnO_2 and PEAI oxidation layer already formed on the surface acts as an effective physical barrier against subsequent oxygen penetration, significantly slowing down the oxidation kinetics. It should be noted that this retardation does not completely halt the oxidation process. Therefore, we have exposed our film to oxygen for an optimal period of time (resulting in the stable device operation while maintaining sufficient bulk material) and the further oxidation was protected through using parylene-c encapsulation. This dual-protection mechanism explains the remarkable stability of our devices, maintaining consistent electrical properties for over a month.

Regarding to the reviewer's suggestion, we have also confirmed this stability trend through time-dependent photoluminescence analysis, as shown in **Supplementary Fig. 10**, which demonstrates similar degradation kinetics. This complementary characterization further supports our understanding of the enhanced device stability.

Page 8, line 18: The same trend was also observed in time-dependent photoluminescence analysis as shown in **Supplementary Fig. 10**.

Supplementary Fig. 10. Time-dependent photoluminescence measurements showing degradation kinetics of

(a) Pe0 and (b) Pe120 with parylene-c encapsulation.

5. The practical application of JFETs using $PEASnI_2$ is unclear. There are several high-performance JFETs based on n-oxides and various p-type semiconductors. A benchmark comparison should be provided.

Reply: We appreciate the suggestion regarding the benchmark comparisons with other JFETs. In response, we have compiled **Revisionz Table 3** that presents a comparison with recently reported JFETs, focusing on 2D material-based devices that constitute the majority of recent high-performance JFET research.

While our device may not exhibit the highest performance in all metrics, it offers significant practical advantages over 2D material-based JFETs. The fundamental limitation of these devices stems from their reliance on mechanical exfoliation of 2D flakes, which presents substantial challenges in terms of both fabrication complexity and precise patterning due to the irregular morphology. Most importantly, this approach makes large scale array fabrication extremely challenging, limiting their further application due to CMOS incompatibility. In contrast, our device employs a solution-processable thin films deposited via simple spin-coating, combined with a mechanical lift-off process that enables patterning. This advantage has allowed us to successfully demonstrate array fabrication and logic gate implementation, representing significant progress toward practical applications. We added these points into the manuscript as follows:

Page 13, line 8: It is worth noting that the successful fabrication of logic gates was enabled by inherent patterability of our device, representing a key advantage over recently reported 2D flake-based JFETs^{28,29,43,47,50,51} which are limited in logic implementation due to their reliance on mechanical exfoliation. This solution-processed approach, combined with controlled oxidation, opens up new practical pathways toward integrated circuits and advances the field of perovskite electronics.

Revision Table 3. Performance comparison between our perovskite b-JFET and recently reported 2D material-based JFETs.

Type	P-type semiconductor	N-type semiconductor	Deposition method (p-type)	Deposition method (n-type)	Mobility ($\text{cm}^2\text{V}^{-1}\text{s}^{-1}$)	Subthreshold swing (mV dec^{-1})	Patternability	Ref
JFET	Ge	MoS ₂	-	Mechanical exfoliation	-	88	NO	[50]
JFET	SnSe	MoS ₂	Mechanical exfoliation	Mechanical exfoliation	27.3	60.3	NO	[29]
JFET	WSe ₂	MoS ₂	Mechanical exfoliation	Mechanical exfoliation	500	100	NO	[47]
JFET	WSe ₂	MoS ₂	Mechanical exfoliation	Mechanical exfoliation	13	200	NO	[47]
JFET	Te	ReS ₂	Drop-casting	Mechanical exfoliation	67.6	229	NO	[28]
JFET	Te	ReS ₂		Mechanical exfoliation	317.6	84	NO	
JFET	p ⁺⁺ Si	MoS ₂	-	Mechanical exfoliation	-	67.4	NO	[51]
JFET	WSe ₂	n ⁺⁺ Si	Mechanical exfoliation	-	171	-	NO	
JFET	GaN	MoS ₂	-	Mechanical exfoliation	-	60.9	NO	[43]
JFET	PEA₂SnI₄	IGZO	Solution spincoating	RF sputtering	29.4	67.1	YES	This work

6. Traditional JFETs typically operate in depletion mode (D-JFET). In a well-designed thin-film pn heterojunction, when the bias applied to the junction is 0V, the width of the depletion layer is usually less than the thickness of the p-type and n-type thin films. For JFETs, this means that when the gate-source voltage is 0V, the device remains always-on because the conductive channel is not pinched off by the depletion region. However, the JFET discussed in this paper operates in enhancement mode (E-JFET). The authors should provide a reasonable explanation for this behavior.

Reply: We appreciate the comment. This is a very important, new aspect of our devices, which we tried to deliver to the reader well. Perhaps, it was done unsatisfyingly. In our device, the naturally formed oxidation

layer (SnO₂:PEAI) on the PEA₂SnI₄ surface plays a crucial role in achieving enhancement mode operation. This oxidation layer creates a type-II staggered band alignment with PEA₂SnI₄, resulting in a significant energy barrier over 1.1 eV at the heterojunction interface. The presence of this large barrier effectively suppresses the gate leakage current at $V_g > 0$.

This behavior is analogous to how wide bandgap materials like GaN can enable enhancement mode operation through their inherent large barrier heights¹. In our case, the oxidation layer serves a similar function - it creates an electron-blocking barrier that prevents gate leakage current flow at positive gate bias; carriers cannot overcome this barrier and form a lateral conductive channel through electron accumulation at the heterojunction interface.

In contrast to conventional JFETs that typically operate only in depletion mode due to insufficient barrier heights at the junction interface, our device successfully achieves enhancement mode operation. This enhancement mode capability is particularly significant for reducing energy consumption, as it ensures near-zero drain current without applied gate voltage. Our detailed band structure analysis and carrier transport studies presented in the manuscript provide strong experimental evidence for this operating mechanism.

To address and emphasize these important points in our manuscript, we included the following sentences in the manuscript.

Page 3, line 26: Furthermore, achieving enhancement mode operation in conventional JFETs is fundamentally limited by large forward bias currents at the PN junction^{43,44}. Therefore, developing JFETs capable of enhancement mode operation while maintaining low leakage current remains a crucial challenge for advancing low-power electronics.

Reference:

- 43 Zhou, Y. *et al.* MoS₂/GaN junction field-effect transistors with ultralow subthreshold swing and high on/off ratio via thickness engineering for logic inverters. *Adv. Funct. Mater.* **34**, 2410954 (2024).
- 44 Kato, Y. *et al.* Dependence of normally-off GaAs JFET performance on device structure. *IEEE Transactions on Electron Devices* **29**, 1755-1760 (1982).

7. *The actual energy band distribution diagram of the device cannot be accurately determined from a simple combination of UPS results for all films. The authors could provide vertical heterojunction IV curves for Pe0-IGZO and Pe120-IGZO devices and extract the Schottky barrier heights from these measurements for comparison. This would make the results more convincing.*

Reply: We appreciate the valuable suggestion. Following the recommendation, we investigated the Schottky barrier height (SBH) at the Au-Pe0 and Au-Pe120 heterojunctions through electrical measurements. We fabricated vertical Schottky diodes and analyzed the current-voltage (I - V) characteristics at various temperatures. The SBH values were extracted using thermionic emission theory, which relates the saturation current (I_{sat}) to the barrier height through the equation $I_{\text{sat}} = AA^*T^2\exp(q\phi_{\text{SB}}/k_{\text{B}}T)$, where A is the area of the Schottky junction, A^* is the effective Richardson constant, q is the elementary charge, k_{B} is Boltzmann's constant, and T is the absolute temperature. By analyzing the temperature-dependent I - V characteristics and plotting $\ln(I_{\text{sat}}/T^2)$ versus $1/k_{\text{B}}T$, we were able to determine the SBH values.

The extracted SBH values were 0.20 eV and 0.91 eV for Pe0-based and Pe120-based Schottky diodes, respectively, which aligns with the UPS measurement. The significant difference in SBH values at the Au-Pe0 and Au-Pe120 interfaces strongly supports our understanding of how air exposure modifies the electronic structure at the perovskite-air interface. We have included the following sentences and **Supplementary Fig. 20** in the manuscript.

Page 10, line 27: To electrically verify the energy barriers determined by UPS, we fabricated vertical Au-Pe0-IGZO-ITO and Au-Pe120-IGZO-ITO Schottky diodes and extracted barrier heights through temperature-dependent I - V measurements (**Supplementary Fig. 20**).

Supplementary Fig. 20. Determination of Schottky barrier heights at Au-Pe0 and Au-Pe120 junctions through thermionic emission analysis. Current-voltage (I - V) characteristics of the (a) Au-Pe0 and (b) Au-Pe120 Schottky junctions under reverse bias conditions. (c) Plots of $\ln(I/T^2)$ versus $1000/T$ of the Schottky diodes

The Schottky barrier heights (SBH) at the Au-Pe0 or Au-Pe120 junctions were investigated using thermionic emission theory⁷. According to this theory, the SBH (ϕ_{SB}) is directly related to the saturation current (I_{sat}) of a Schottky diode through the following equation:

$$I_{sat} = AA^*T^2 \exp(q\phi_{SB}/k_B T), \dots\dots\dots (eq.1)$$

where A is the area of the Schottky junction, A^* is the effective Richardson constant, q is the elementary charge, k_B is Boltzmann constant, and T is the absolute temperature. I_{sat} value was determined from the y -intercept of the extrapolated lines under reverse bias conditions ($V < 0$), as shown in **Supplementary Figs. 20a** and **b**). Subsequently, ϕ_{SB} of for each Schottky diode was extracted from the slope of the $\ln(I_{sat}/T^2)$ versus $1/k_B T$ plot (**Supplementary Fig. 20c**). Analysis revealed SBH values of 0.20 eV and 0.91 eV for the Pe0-based and Pe120-based Schottky diodes, respectively. These results showed align with the UPS measurements, validating our theoretical approach.

References

7. Choi. Y. J. et al. Remote gating of schottky barrier for transistors and their vertical integration. *ACS Nano*. **13**, 7, 7877-7885 (2019)

8. *The carrier concentrations and mobilities of IGZO and PEASnI2 thin films measured by Hall effect should be given, which is convenient for comparison with device characteristic parameters.*

Reply: We appreciate the comment. In **Supplementary Fig. 5**, we have already provided the Hall effect measurement of IGZO thin films. The measurements were performed using a hall bar device and have included both the optical microscope image of the fabricated hall bar device and the measured Hall voltage as a function of applied magnetic field. From these measurements, we extracted an electron carrier density that was essential for JFET mobility calculation presented in our work. For PEA₂SnI₄ thin films, Hall measurements were not conducted due to the high susceptibility to oxidation of the material under ambient conditions, which makes reliable Hall measurements extremely challenging.

9. *In Fig. 5d, the author should give the time scale of the abscissa axis in order to evaluate the operating frequency of the device*

Reply: We appreciate the comment. As the reviewer suggested, we added the time scale to the abscissa axis in **Figure 5d**.

Figure 5. Fabrication of logic gates. (a) OM image and corresponding circuit diagram of a logic circuit for the NOT gate. (b) Voltage-transfer (left) and gain (right) characteristics of the NOT gate under various V_{DD} . (c) OM images and corresponding circuit diagrams of logic gates for the NAND (top) and NOR (bottom) gates at $V_{DD} = 1$ V. (d) Four possible logic combinations (0,0), (1,0), (0,1), and (1,1) with the corresponding output voltages of the NAND (top) and NOR (bottom) gates.

Reviewer #3

The manuscript “Revisiting the role of oxidation in stable and high-performance lead-free perovskite-IGZO junction field effect transistors” by Kim et al., reported on the beneficial effect of Sn oxidation in suppressing the leakage current in the so-called barriered-JFET. Although this work was done with care, there are some fundamental issues stopping me from supporting its publication:

1. The premise of constructing a bi-layer device must be that the resultant performance is much better than the individuals, which justifies the more complex structure. However, this comparison is absent in the present study despite the vast amount of literature on both Sn-based hybrid perovskite and IGZO transistors. In fact, it is puzzling to me why and how the authors conceived the idea of making this kind of bi-layer junction device in the first place. Is it aimed at presenting the Sn oxidization? There are many ways to achieve that goal like encapsulation, designing a bilayer structure with perovskite at the bottom, and so on.

Reply: We appreciate the reviewer for raising a fundamental question about our bilayer device structure. Our research primarily focused on transforming what has traditionally been considered a critical weakness of Sn-based perovskites – their susceptibility to oxidation – into an advantageous feature for device performance.

To address performance comparisons, we have expanded our introduction to include the references from **Revision Tables 1 and 2**. As demonstrated in **Revision Table 1**, Sn-based perovskites exhibit promising potential as channel materials due to their high mobility. However, their extreme sensitivity to air remains a critical challenge, with devices becoming non-functional within minutes of exposure. Even with various preventive strategies, including encapsulation, achieving stable operation has proven elusive.

Regarding the reviewer's question about our motivation for developing this bilayer junction device, we deliberately utilized the oxidation phenomena rather than preventing it, to achieve better performing electronic devices. Through a simple two-hour thermal treatment at 80°C in the air, a natural oxide layer on the perovskite surface was created, which serves as an effective barrier for i) leakage current to the gate terminal ii) oxygen adsorption at the surface. This straightforward process not only ensured device stability but also improved device performance, including lower subthreshold voltage, operating voltage, and carrier mobilities.

While some IGZO-FETs using high- k dielectrics have achieved comparable or superior performance metrics (as shown in **Revision Table 2**), these devices typically rely on expensive, complex, and high-temperature ALD processes for dielectric layer formation. In contrast, our solution-processed, low-temperature approach offers significant advantages in manufacturing simplicity and cost-effectiveness.

We have completely modified the introduction to include this discussion and references from **Revision Tables 1 and 2** to help readers better understand the key points of our research.

Introduction:

Organometallic halide perovskites have gained significant attention owing to their tunable bandgap, facile processability, and high carrier mobility¹⁻³. These properties have enabled major advances in optoelectronics, particularly in solar cells and light-emitting diodes, while also opening new possibilities for transistor applications⁴⁻¹¹. Although Pb-based perovskites have demonstrated the most excellent optoelectronic properties, their toxicity poses serious environmental concerns and limits industrial applications¹²⁻¹⁴. Sn-based perovskites have emerged as promising alternatives due to their similar valence electron configuration^{11,15-17}. However, susceptibility of Sn to oxidation due to the absence of the lanthanide shrinkage effect, transitioning from Sn²⁺ to Sn⁴⁺, has hindered their practical implementation¹⁸⁻²⁰. Previous research has shown that Sn-based perovskites demonstrate highly promising performance as the p -type channel material in thin-film transistors; however, even the slightest exposure to air within mere minutes leads to self- p -doping, which results in metallic characteristics that hinder their function as transistors^{4,11,21-25}. While numerous research efforts have focused on suppressing this oxidation through various chemical and physical strategies, these approaches face inherent limitations^{26,27}. Considering the spontaneity of Sn oxidation and distinctive evolution in its electronic structures, the unique electronic properties of oxidized Sn-based perovskites could potentially create new opportunities in electronic devices.

The increasing prevalence of Internet of Things (IoT) devices has made low power consumption a critical requirement in modern electronics. Junction field-effect transistors (JFETs) have emerged as a promising candidate for such applications due to their steep subthreshold swing values^{28,29}. Unlike

conventional metal-oxide-semiconductor field-effect transistors (MOSFETs), which require sophisticated dielectric engineering to approach the theoretical limit of 60 mV dec^{-1} at 300K^{30-42} , JFETs can theoretically achieve this performance more readily due to their inherently high gate capacitance achieved without a gate dielectric layer²⁹. However, this architecture presents two significant challenges: substantial gate leakage current and operation limited to depletion mode. The depletion-mode constraint is particularly problematic for power consumption since these devices conduct significant current at zero gate voltage. Furthermore, achieving enhancement mode operation in conventional JFETs is fundamentally limited by large forward bias currents at the PN junction^{43,44}. Therefore, developing JFETs capable of enhancement mode operation while maintaining low leakage current remains a crucial challenge for advancing low-power electronics.

In this study, we successfully fabricated a stable and high-performance transistor by tailoring and exploiting the oxidation phenomenon of Sn-based perovskites. Specifically, a JFET was fabricated with *p*-type PEA_2SnI_4 and *n*-type indium gallium zinc oxide (IGZO), which formed a PN junction. While the as-prepared JFET exhibited poor performance due to significant gate leakage current, the JFET based on the perovskite layer with an oxidized surface exhibited dramatic changes in its operation. By exploiting the oxidized layer instead of preventing its formation, the gate leakage current of the JFET was suppressed below 10^{-10} A . Moreover, the JFET was able to operate unconventionally in the enhancement mode owing to the suppression of gate leakage current by the oxidation layer, thereby overcoming one of the fundamental limiting factors of conventional JFETs. We refer to such a device as a barriered JFET (b-JFET). Owing to the high polarizability of its perovskite layer, the resulting perovskite b-JFET achieved an averaged field-effect mobility of $29.4 \text{ cm}^2\text{V}^{-1}\text{s}^{-1}$, a low subthreshold swing (SS) of 67.1 mV dec^{-1} , and a high on/off current ratio of $>10^5$ at a voltage of $\leq 1 \text{ V}$. Moreover, the device exhibited excellent bias stress stability, operational stability, and environmental stability over a month in ambient air. Additionally, we successfully constructed logic circuits and achieved high inverter gain values, further confirming the potential of the device for various practical applications.

Revision Table 1. Performance comparison between the perovskite-employed field effect transistors.

Employed as	Composition	Dielectric	Mobility ($\text{cm}^2 \text{V}^{-1} \text{s}^{-1}$)	Operation voltage (V)	Air stability	Ref
Channel	FACsPEASnI	SiO_2	17.4	-40 to 40	-	[21]
Channel	CsFAPEASnI	SiO_2	70	-40 to 40	2 min*	[22]
Channel	PEAFASnI	PMMA- Al_2O_3	0.21	-15 to 5	-	[11]
Channel	CsSnI_3	SiO_2	50	-40 to 40	1 min*, 36 hours	[23]
Channel	FPEASnI	SiO_2	2.96	-40 to 10	30 min	[4]
Channel	FAPEASnI	HfO_2	12	-8 to 8	-	[24]
Channel	$(\text{BA})_2\text{SnI}_4$	PVA/CL-PVP	0.03	-60 to 20	-	[25]
Gate	PEA_2SnI_4	-	29.4	-1 to 1	1 month	This work

* *unencapsulated*

Revision Table 2. Performance comparison between the solution processed perovskite-IGZO JFET and previously reported IGZO-based transistors with high-k dielectrics.

Type	Dielectric (perovskite) deposition method	Channel material	Subthreshold swing (mV dec^{-1})	Process temperature ($^\circ\text{C}$)	Mobility ($\text{cm}^2 \text{V}^{-1} \text{s}^{-1}$)	On/off ratio	Operating voltage (V)	Ref
MOSFET	ALD	IGZO	62.29	400	18.9	2.8×10^9	-1 to 3	[30]
	ALD	IGZO	70.2	250	55.3	$\sim 10^9$	-1 to 3	[31]
	PEALD	IGZO	76	350	6.1	3.5×10^9	-3 to 10	[32]
	ALD	IGZO	87	300	19.7	4.3×10^{11}	-1 to 4	[33]
	RF sputtering	IGZO	88	150	2.3	5×10^5	0 to 1	[34]
	ALD	IGZO	90	150	10.4	2.8×10^9	-5 to 5	[35]

	PEALD	IGZO	130	250	21.7	3.2×10^8	-2 to 2	[36]
	Solution spin-coating	IGZO	140	350	85	$\sim 10^4$	-2 to 2	[37]
	ALD	IGZO	170	150	15.1	-	-5 to 5	[38]
	E-beam	IGZO	250	RT	61.5	$\sim 10^5$	-2 to 3	[39]
	ALD	IGZO	256	150	9.7	1.3×10^6	-2 to 10	[40]
	Printing	IGZO	300	150	4.3	10^8	-4 to 4	[41]
	RF sputtering	IGZO	560	RT	28	10^7	-6 to 10	[42]
JFET	Solution spin-coating	IGZO	67.1	100	29.4	1.2×10^5	-1 to 1	This work

2. The thoughts on the device structure lead me to other doubts about the use of a natural oxidization layer on Sn perovskite as the dielectric layer. First, it is not surprising at all that the leakage current is high with the top gate since there is nothing there to stop it. Why not compare with the bottom gate? Second, such a natural oxidization layer will not be a good dielectric layer, much inferior to conventional ones such as SiO_2 and alumina. Why bother? In the discussion, the associated capacitance was mentioned in passing to be high, but it was not directly characterized. Why? If this thin layer is so important, it should be characterized carefully, for example using TEM to check the uniformity and crystallinity, and control experiments to check factors like oxidization temperature and duration etc.

3. in the line of scrutinizing the role of the natural oxidization layer, why not use a thin dielectric layer like PMMA or others to suppress the leakage layer? Does the perovskite layer really play any role at all? I doubt the oxidized layer brings in anything magic like ultra-high dielectric constant, super-high resistivity, or something like it.

Reply for comment 2 and 3: We appreciate the comments regarding our device structure and characterization methodology. We would like to address several critical aspects of our work through comprehensive analysis of both the role of oxidation layer and alternative approaches.

First, it is crucial to clarify that the natural oxidation layer of Sn-based perovskite does not function as a dielectric layer. In **Supplementary Fig. 6a-c**, capacitance (C)-frequency (f) characteristics of Au-perovskite-ITO structure is demonstrated. Not only in Pe0, but also in the Pe120-based Au-perovskite-ITO structure, the phase angle converges to 0 degrees despite the presence of oxidation layer. This indicates that the oxidation layer forms a mere resistive barrier rather than acting as a dielectric. In contrast, the device incorporating IGZO (**Supplementary Fig. 6d-f**) exhibits distinctly different behavior, characterized by a non-zero phase angle. This observation aligns with the semicircle observed in the Nyquist plot (**Supplementary Fig. 17**). The semicircle observed at intermediate frequency (10 - 10^4 Hz) region can be attributed to charge accumulation at the perovskite-IGZO interface¹. The enlarged semicircle in the Pe120-based device indicates an increased resistance between IGZO and perovskite, possibly due to the surface layer formed through oxidation. These electrical characteristics are fundamental to the operation of the device and support our classification of the device as a JFET rather than a MOSFET.

Page 7, line 2: Note that the mobility calculation does not rely on the capacitance measurements, as we employ the JFET mobility calculation equation rather than the conventional MOSFET equation. This classification of our device as a JFET is supported by our impedance analysis (**Supplementary Fig. 6**), which shows that neither the perovskite layer nor the perovskite-oxide layer functions as a conventional dielectric.

Supplementary Fig. 6. Frequency-dependent capacitance and phase angle measurements of perovskite devices. (a) Schematic of Au-perovskite-ITO device structure. (b), (c) Bode plots showing the capacitance and phase angle *versus* frequency for Pe0 and Pe120, respectively. (d) Schematic of Au-perovskite-IGZO-ITO device structure (e-f) Bode plots showing the capacitance and phase angle *versus* frequency for Pe0 and Pe120, respectively.

Analysis of the Bode plots demonstrates fundamental differences in device behavior with and without IGZO integration. The phase angle convergence to 0 degrees in devices without IGZO (**Supplementary Fig. 6a-c**) indicates purely resistive characteristics. Conversely, IGZO-incorporated devices (**Supplementary Fig. 6d-f**) exhibit capacitive behavior with a non-zero phase angle, correlating with the semicircular response in the Nyquist plot (**Supplementary Fig. 17**). The observed semicircle in the intermediate frequency range (10-10⁴ Hz) indicates charge accumulation at the perovskite-IGZO interface¹. Based on these electrical characteristics, we classified our device as a JFET and accordingly employed a capacitance-independent mobility calculation method to ensure accurate mobility assessment.

References

1. Abdulrahim, S. M., Ahmad, Z., Bahadra, J. & Al-Thani, N. J. Electrochemical impedance spectroscopy analysis of hole transporting material free mesoporous and planar perovskite solar cells. *Nanomaterials* **10**, 1635 (2020).

As the reviewer suggested in comment 3, we investigated whether similar effects could be achieved by depositing a thin dielectric layer on top of the perovskite instead of using an oxidation layer (**Supplementary Fig. 18**). While devices incorporating PMMA instead of the oxidation layer could be operated in a similar manner to those described in this manuscript, this approach revealed several significant limitations. First, the thin PMMA layer not only proved ineffective at suppressing leakage current but also, due to its low-*k* characteristics, resulted in substantially degraded subthreshold swing performance compared to our oxidation approach. The fabrication process was also more complex, as our natural oxidation layer forms through a simple thermal treatment in ambient air, while the PMMA approach requires additional deposition steps. Most notably, we observed a crucial difference in hysteresis characteristics. The natural oxidation layer forms a homogeneous connection with the perovskite layer, resulting in minimal interface traps and consequently negligible hysteresis. In contrast, the heterogeneous junction between PMMA and perovskite creates interface traps that lead to significant hysteresis. This distinction highlights a fundamental advantage of our natural oxidation approach over conventional dielectric layers. These findings demonstrate that the natural oxidation layer not only simplifies device fabrication but also provides unique benefits in terms

of interface quality and device performance that cannot be readily achieved with conventional dielectric materials like PMMA.

Supplementary Fig. 18. (a) Device structure of Pe120-based perovskite b-JFET (left panel) and corresponding transfer characteristics (right panel). (b) Device structure of PMMA incorporated Pe0-based perovskite b-JFET (left panel) and corresponding transfer characteristics (right panel).

Regarding the concern about the oxidation layer quality, we acknowledge that naturally formed oxidation layers may be less uniform than precisely deposited dielectric films. Following this suggestion, we conducted cross-sectional transmission electron microscopy (TEM) analysis to examine the microscale uniformity and crystallinity of the films (**Revision Fig. 1**). Unfortunately, we were unable to directly visualize the surface layer structure. This is particularly because we had to use the focused-ion-beam (FIB) technique for sample preparation, where the high-energy ion beam may have caused structural damage such as amorphization or ion implantation, disrupting the crystal structure of perovskite films (both bulk and oxidized surface). Moreover, the Pt coating applied to protect the perovskite surface may have diffused into the surface layer of the film, obscuring the thin bilayer structure and making it difficult to distinguish in TEM imaging.

Revision Fig. 1. TEM and EDS elemental mapping.

Regarding the impact of oxidation parameters on film quality, while direct microscale observation using the aforementioned methods was not feasible due to practical constraints of cost and time, we were able to indirectly elucidate film formation and quality through electrical measurements. The effects of oxidation time are extensively explored in the main text, while the temperature-dependent transfer characteristics are presented in **Supplementary Fig. 22**. Our findings demonstrate that oxidation temperature predominantly influences the reaction kinetics. Notably, temperatures above 100°C seem to exceed a critical threshold that compromises the structural stability of PEA₂SnI₄, leading to degradation of the perovskite structure.

The rationale behind our device architecture stems from our goal to achieve minimal subthreshold swing by leveraging the perovskite layer's inherently high capacitance, while utilizing the oxidation layer as an electron barrier rather than a conventional dielectric medium. We have included these additional analyses and supporting figures in the manuscript.

Page 11, line 8: The transfer characteristics of perovskite b-JFETs as a function of annealing temperature are presented in **Supplementary Fig. 22**.

Supplementary Fig. 22. Transfer characteristics of perovskite b-JFETs as a function of annealing temperature.

Page 10, line 4: To scrutinize the role of this surface layer, we compared the transfer characteristics with a device incorporating a thin PMMA layer designed to serve a similar function as the surface oxidation layer (**Supplementary Fig. 18**). While the PMMA-based devices could be turned on, they exhibited higher leakage current and degraded subthreshold swing due to the lower dielectric constant of PMMA compared to the surface layer. More significantly, we observed large hysteresis in the transfer curve, which can be attributed to interface traps arising from the heterogeneous junction between PMMA and the perovskite layer. In contrast, our surface layer forms a homogeneous junction with the perovskite, minimizing interface traps and resulting in significantly reduced hysteresis in the transfer characteristics.

Reviewer #2 (Remarks to the Author):

The authors have addressed my concerns properly. Please see below as my comments on authors' response to Reviewer #1's concerns.

Comment 1: "The authors report on perovskite-IGZO junction field-effect transistors. Although some results are interesting, in my personal opinion, this manuscript does not meet the standard of Nature Communications.

1. The proposed perovskite-IGZO FET demonstrated n-type behavior with a field-effect mobility of $29.4 \text{ cm}^2\text{V}^{-1}\text{s}^{-1}$ and an on/off current ratio of $>10^5$. However, this device performance should be easily realized by IGZO FET with much simpler process. In addition, the logic gates should also be realized by the IGZO FET."

Opinion: The authors restate their perovskite-IGZO JFET figure of merits (low operation voltage = 1 V, SS = 67.1 mV/dec, inverter gain = 27.4, 100 °C processing) and highlight the scalable manufacturing processes. As far as I know, high-performance IGZO TFTs with low voltage operation (including mobility over $20 \text{ cm}^2/\text{V s}$, SS value close to 60 mV/decade, and operating voltage as low as 1 V) can be realized by high-quality high- κ dielectric layers such as ALD and anodic oxidation, and the deposition temperature can also be kept low, at least similar to $\sim 100^\circ\text{C}$ proposed by this work. The scalability and manufacturability of these methods are significantly higher than that of JFETs formed by solution-processed perovskite and IGZO proposed by the authors. In addition, the gain of pseudo-CMOS inverter composed of pure IGZO TFTs can easily exceed 50, and the performance of the inverter is not outstanding enough.

Recommendation: The authors should re-examine the real advantages of perovskite/IGZO JFETs and the scientific motivation behind this work. The authors should point out at least one or two unique advantages of their devices, which are not available in all other IGZO TFT with high performance and low voltage. In addition, the authors should provide a side-by-side table comparing their inverter (V_{DD} , gain, SS, static power, processing temperature) with representative IGZO TFT based inverter from the literature. It is well known that IGZO technology exhibits excellent scalability, and the authors should prove that the IGZO JFETs prepared by the method proposed in this work did not lead to the loss of IGZO scalability and large-scale manufacturing. As a result, the authors should provide device yield and uniformity metrics (evaluated 100 devices on a 4-inch wafer) to justify the proposed "scalability".

Reply: We deeply appreciate the constructive and thoughtful feedback and recommendations by the reviewer. In response, we have carefully revised the Introduction to clearly articulate the unique advantages of our device over conventional IGZO TFTs and other advanced transistor architectures. Specifically, we emphasize the following key advantages:

1. Enhancement-mode JFET operation via a simple, solution-processable approach.

Conventional JFETs are inherently limited by large gate leakage, and they are operated under depletion-mode only, making them less suitable for low-power applications. In contrast, our b-JFET overcomes these limitations through controlled surface oxidation of the Sn-based perovskite, enabling enhancement-mode operation with suppressed gate leakage below 10^{-10}A . Notably, this is achieved without relying on epitaxial growth, doping engineering, or vacuum-deposited gate dielectrics—processes that are typically essential in MOSFETs or HEMTs to realize comparable functionality.

2. Steep subthreshold swing enabled by favorable heterojunction interface properties.

Our device achieves an SS of 67.1 mV dec^{-1} , approaching the theoretical limit at room temperature. This is attributed not only to the high intrinsic gate capacitance of the perovskite layer (a key characteristic of our device, as already highlighted in the main text), but also to the low interface trap density at the perovskite/IGZO junction. We cite recent work (Hu et al., *Adv. Mater.*, 2025) demonstrating that perovskite/inorganic oxide buried interfaces (e.g., perovskite/ SnO_2) exhibit reduced trap-assisted recombination and efficient carrier extraction compared to the perovskite/organic hole transport layer (HTL) interfaces. Given that IGZO belongs to the same class of wide-bandgap metal oxides as SnO_2 , we suggest that our heterointerface similarly benefits from minimized interfacial traps, contributing to the excellent switching characteristics observed. These additions are reflected in the Introduction and are also discussed in the main text (Page 9, line 2).

In addition to these device-level advantages, we believe our work provides important scientific insights with broader implications. As mentioned in the first paragraph in the Introduction, lead-free perovskites are being actively explored as a key direction toward the practical realization of perovskite-based

electronics. However, the severe oxidation of tin has posed a major obstacle to their use in electronic devices. In this study, rather than suppressing oxidation, we uniquely exploited the oxidation process itself as a functional mechanism to improve device performance. To the best of our knowledge, this is the first report demonstrating that the controlled oxidation of Sn-based perovskites can be utilized to engineer a barrier layer that enhances the

Revised introduction:

Organometallic halide perovskites have gained significant attention owing to their tunable bandgap, facile processability, and high carrier mobility¹⁻³. These properties have enabled major advances in optoelectronics, particularly in solar cells and light-emitting diodes, while also opening new possibilities for transistor applications⁴⁻¹¹. Although Pb-based perovskites have demonstrated the most excellent optoelectronic properties, their toxicity poses serious environmental concerns and limits industrial applications¹²⁻¹⁴. Sn-based perovskites have emerged as promising alternatives due to their similar valence electron configuration^{11,15-17}. However, susceptibility of Sn to oxidation due to the absence of the lanthanide shrinkage effect, transitioning from Sn²⁺ to Sn⁴⁺, has hindered their practical implementation¹⁸⁻²⁰. Previous research has shown that Sn-based perovskites demonstrate highly promising performance as the p-type channel material in thin-film transistors; however, even the slightest exposure to air within mere minutes leads to self-p-doping, which results in metallic characteristics that hinder their function as transistors^{4,11,21-25}. While numerous research efforts have focused on suppressing this oxidation through various chemical and physical strategies, these approaches face inherent limitations^{26,27}. Considering the spontaneity of Sn oxidation and distinctive evolution in its electronic structures, the unique electronic properties of oxidized Sn-based perovskites could potentially create new opportunities in electronic devices.

The increasing prevalence of Internet of Things (IoT) devices has made low power consumption a critical requirement in modern electronics. Junction field-effect transistors (JFETs) have emerged as a promising candidate for such applications due to their steep subthreshold swing (SS) values^{28,29}. Unlike conventional metal-oxide-semiconductor field-effect transistors (MOSFETs), which often require advanced techniques (such as vacuum-deposited ultra-thin high-k gate dielectrics) and complex interface engineering³⁰⁻⁴² to achieve optimal performance, JFETs can theoretically achieve near-ideal SS without a gate dielectric layer owing to their inherently high capacitance²⁹. However, this architecture presents two significant challenges: substantial gate leakage current and operation limited to depletion mode. The depletion-mode constraint is particularly problematic for power consumption since these devices conduct significant current at zero gate voltage. Furthermore, achieving enhancement mode operation in conventional JFETs is fundamentally limited by large forward bias currents at the PN junction^{43,44}. Due to these inherent limitations, research into JFET technologies has made limited advancement in recent years, while alternative transistor architectures such as MOSFET and high electron mobility transistors (HEMTs) have gained increasing prominence. Yet, these enhancement-mode operating devices are realized at the expense of increased fabrication complexity. For instance, MOSFETs often require gate stack dipole engineering, while HEMTs typically require epitaxially grown heterostructures and precise barrier/channel band alignment strategies⁴⁵. Therefore, developing JFETs capable of enhancement mode operation while maintaining low leakage current remains a crucial challenge for advancing low-power electronics.

In this study, we report a simple, low-temperature, solution-processable, scalable strategy to fabricate high-performance enhancement mode JFETs by tailoring and leveraging the oxidation phenomenon of Sn-based perovskites. Specifically, we formed a PN junction using p-type PEA₂SnI₄ and n-type indium gallium zinc oxide (IGZO). While the as-fabricated device suffered from large gate leakage, we turned the above-mentioned conventional drawback of Sn-based perovskites (i.e., their susceptibility to oxidation) into a functional advantage. Controlled surface oxidation of the Sn-based perovskite resulted in the formation of a barrier layer that effectively suppressed gate leakage to below 10⁻¹⁰ A. This enabled enhancement-mode operation, overcoming a key limitation of conventional JFETs. We refer to this architecture as a barriered JFET (b-JFET). Our b-JFET achieved excellent electrical properties, including a field-effect mobility of 29.4 cm²V⁻¹s⁻¹, a low SS of 67.1 mV dec⁻¹, and an on/off current ratio exceeding 10⁵ under ≤1 V operation. Moreover, we demonstrated robust environmental, bias, and operational stability over extended periods in ambient air, despite containing typically unstable Sn-perovskite layer, highlighting the robustness of our approach. Additionally, we successfully constructed logic circuits and

achieved high inverter gain values with low applied voltages, further confirming the potential of the device for various practical applications.

Page 8, line 5: Furthermore, the low interface trap density at the perovskite/IGZO junction likely contributes to the steep SS observed. Recent studies have shown that perovskite/inorganic oxide buried interfaces, such as those with SnO₂, exhibit reduced trap-assisted recombination and efficient carrier extraction, which supports the favorable interfacial electronic properties of such heterojunctions⁴⁹.

Reference:

- 45 Roccaforte, F., Greco, G., Fiorenza, P. & Iucolano, F. An overview of normally-off GaN-based high electron mobility transistors. *Materials* **12**, 1599 (2019).
- 49 Hu, B. *et al.* Revealing trapped carrier dynamics at buried interfaces in perovskite solar cells via infrared-modulated action spectroscopy with surface photovoltage detection. *Adv. Mater.* doi: 10.1002/adma.202502160.

To address the benchmark comparison of NMOS inverters, we have included **Supplementary Table 2**, which quantitatively compares representative IGZO FET-based inverters employing solution-processed dielectrics. We focused on solution-processed devices for comparison, as they offer advantages of low-cost fabrication processes. Despite generally lower performance than vacuum-processed counterparts, solution-processed transistors are in increasing interest for large-area, low-power, and cost-sensitive applications, making them commercially relevant for future electronic systems. Our device demonstrates the highest voltage gain at $V_{DD} = 1$ V and the lowest processing temperature among the reported solution-processed counterparts. Unfortunately, static power was not mentioned in most of the literature, precluding a direct comparison.

Supplementary Table 2. Benchmark comparison of IGZO FET-based inverters with solution-processed gate dielectrics

Gate dielectric	Deposition method	Inverter gain	V_{DD}	Gain at $V_{DD} = 1$ V	Process temp. (°C)	Pull-up device	SS (mV/dec)	Ref
HfGdO _x	Spin-coating	19.8	5	3.9	450	resistor	70	[1]
PVP-co-PMMA/Al ₂ O ₃	Spin-coating	17.3	3	5.7*	300	PMOS	-	[2]
ZrO ₂	Spin-coating	10.8	5	2.2*	200	NMOS	300	[3]
ZAO	Spray-coating	58	6	9.7*	420	NMOS	115	[4]
HfAlO	Spin-coating	4.46	4	0.9	450	NMOS	87	[5]
Al ₂ O ₃	Spin-coating	32.2	5	6.4*	-	NMOS	-	[6]
HfLaOx	Spin-coating	7.59	2.5	3.0*	300	resistor	140	[7]
Perovskite**	Spin-coating	27.4	1	27.4	100	NMOS	67.1	This work

* Estimated value derived by normalizing the reported inverter gain to a V_{DD} of 1 V.

** Not a gate dielectric, but a p-type gate semiconductor in JFET.

References

- Zhang, Y., *et al.* Aqueous-solution-driven HfGdO_x gate dielectrics for low-voltage-operated α -InGaZnO transistors and inverter circuits. *J. Mater. Sci. Technol.* **50**, 1-12 (2020).
- Cho, H. J., *et al.* Solution-processed organic-inorganic hybrid gate insulator for complementary thin film transistor logic circuits. *Th. Sol. Films* **673**, 14-18 (2019).
- Park, S.-J., Ha, T.-J. Sol-gel-based metal-oxide thin-film transistors for high-performance flexible NMOS inverters. *J. of Alloys Compd.* **912**, 165228 (2022).
- Islam, M. M., *et al.* Spray-pyrolyzed high-k zirconium-aluminum-oxide dielectric for high performance metal-oxide thin-film transistors for low power displays. *Adv. Mater. Interfaces* **8**, 2100600 (2021).
- He, G., Li, W., Sun, Z., Zhang, M., Chen, X. Potential solution-induced HfAlO dielectrics and their applications in low-voltage-operating transistors and high-gain inverters. *RSC Adv.* **8**, 36584-36595 (2018).
- Park, S. J., Ha, T. J. Microwave-irradiated metal-oxide thin-film transistors with recessed gate structure and their applications in logic circuits. *IEEE Trans. Electron Dev.* **70**, 99-104 (2023).

7 Wang, W., He, G., Wang, L., Xu, X., Zhang, Y. Solution-driven HfLaO_x-based gate dielectrics for thin film transistors and unipolar inverters. *IEEE Trans. Electron Dev.* **68**, 4437-4443 (2021).

The static power consumption (P_{static}) of our inverter can be estimated using the relation: $P_{\text{static}} = V_{\text{DD}} \times I_{\text{static}}$. In our NMOS inverter, static power primarily arises from leakage current through the pull-up JFET when it is in the off state (i.e., $V_{\text{GS}} = 0$ V). From the transfer curves of a hundred JFETs in Supplementary Fig. 7, the averaged drain current under this condition is approximately 6.6×10^{-10} A. Given $V_{\text{DD}} = 1$ V, the corresponding P_{static} is approximately 660 pW. This remarkably low static power is attributed to the enhancement mode operation of the b-JFET, ensuring near-zero current flow in the standby state. This point has been emphasized in the revised main text.

Page 13, line 2: In addition, the inverter exhibited low static power consumption of approximately 660 pW at $V_{\text{DD}} = 1$ V, owing to the enhancement mode operation of the b-JFET that ensures negligible standby current (averaged $I_{\text{static}} \approx 6.6 \times 10^{-10}$ A when $V_{\text{IN}} = 0$ V).

To address the concern of reviewer regarding the scalability, we fabricated a 4-inch wafer-scale b-JFET array to provide device yield and uniformity metrics evaluated on a hundred devices (revised **Figure 2a** and **2d**, Supplementary Fig. 7).

Figure 2. Electrical performances of perovskite b-JFETs (a) Image of the two-inch wafer scale of the perovskite b-JFETs.

Comment 2: “The advantages of the perovskite–IGZO JFETs have not been highlighted”

Opinion: Although the introduction was rewritten to list features (low leakage, enhancement-mode, high μ /on–off, stability, logic demo), same as Comment 1, the author still has not put forward the advantages of IGZO JFET fundamentally, and the high mobility and low SS value can still be achieved by industrial compatible and high reliability ALD technology. Moreover, I do not agree with what the author claimed “sophisticated dielectric engineering”.

Recommendation: Same as Comment 1.

Reply: We thank the reviewer for the feedback. As noted in our response to Comment 1, we have revised the Introduction to clearly articulate the distinct advantages of our b-JFET over conventional IGZO-based transistors. In addition, in response to the reviewer’s concern regarding the phrase “sophisticated dielectric engineering”, we have revised the expression to a more technically specific and objective description. In the revised Introduction, we now refer to “advanced techniques such as vacuum-deposited ultra-thin high-k gate dielectrics and complex interface engineering.” This phrasing was selected to avoid subjective language, while still capturing the fact that such fabrication steps (although widely used) require additional equipment, process control, and fabrication cost compared to gate-free, solution-processed b-JFET approach. By replacing “sophisticated” with “advanced” and specifying the techniques involved, we believe the revised wording is both technically accurate and appropriately neutral in tone.

Introduction: ~ which often require advanced techniques (such as vacuum-deposited ultra-thin high-k gate dielectrics) and complex interface engineering³⁰⁻⁴² to achieve optimal performance, JFETs can theoretically achieve near-ideal SS without a gate dielectric layer owing to their inherently high capacitance²⁹.

Comment 3: Figure 2b: transfer curve at lower V_{DS} (< 0.1 V)?

Opinion: The authors added Supplementary Fig. 4 with different V_{DS} values, which is good. But why the current decrease over 3 orders of magnitude when the $V_{DS} = 0.01$ V (only decrease 2 orders of magnitude compared with $V_{DS} = 1$ V). Does this mean that the IGZO JFETs is a non-ideal device?

Recommendation: Authors should provide proper explanation of the current collapse at $V_{DS} = 0.01$ V.

Reply: We appreciate the reviewer's insightful observation regarding the transfer characteristics at low drain voltage ($V_{DS} = 0.01$ V), as presented in Supplementary Fig. 4. We acknowledge that, under this condition, the drain current (I_{DS}) exhibits a noticeable suppression—more than 2 orders of magnitude lower compared to the case with $V_{DS} = 1$ V. We attribute this behavior to the influence of gate-induced leakage currents, which become significant under low drain bias.

As shown in, the gate leakage current increases to the nanoampere (nA) level as V_{GS} approaches 0.5 V. At $V_{DS} = 0.01$ V, the intrinsic on-current of the device is also in the nA range, resulting in a situation where a substantial portion of the injected carriers is diverted through the gate leakage path rather than contributing to source–drain conduction. This effect is intrinsic to the geometry and operating principle of JFETs, especially under conditions where the gate potential greatly exceeds the drain bias ($V_{GS} \gg V_{DS}$).

Supplementary Fig. 4. Transfer characteristics of the perovskite-b-JFET measured at different drain voltages ($V_{DS} = 1, 0.5, 0.1, 0.05$ and 0.01 V).

At $V_{DS} = 0.01$ V, I_{DS} initially increases with V_{GS} , but shows a decline beyond ~ 0.4 V. This is due to gate-induced leakage current, which becomes comparable to the on-current at low drain bias. The gate leakage reaches the nA range at $V_{GS} = 0.5$ V, effectively reducing net drain current. This effect diminishes at higher V_{DS} , where the on-current dominates.

Comment 4: P7 line 1, how about the capacitance at low frequency (<20 Hz)? Please show the capacitance-frequency curve. Besides, the device mobility should be calculated at low frequency to avoid mobility overestimation.

Opinion: The reply discusses JFET classification and a new mobility formula, but omits crucial details on the explicit frequency range measured (< 20 Hz), the numerical values used (N_s , g_m , t), and a direct comparison of μ extracted via low-frequency capacitance versus the JFET formula. In addition, I think it is biased to use this formula $\mu = (L/W)(g_m)/(q \cdot N_s \cdot t)$ to calculate the mobility of JFETs. This is because, N_s presented the electron concentration in the formula is obtained by Hall effect test. As we all know, the carrier concentration test result of Hall effect is influenced by many factors (such as the contact resistance of Hall electrodes, the intensity of Hall magnetic field, etc.), and the result cannot accurately reflect the absolute carrier concentration values of IGZO thin films. From the formula, when the value of N_s is doubly underrated, the mobility of the device is doubly overrated. In Hall test, it is very common that the value of N_s changes by more than one order of magnitude under different test conditions of the same sample. Thus, this method of extracting mobility should be treated with great caution.

Recommendation: The author should use three different methods to evaluate the device mobility, and compare the difference. If the difference is not so big, it can be considered that the mobility extraction is accurate.

Firstly, the authors should recalculate the mobility using conventional C-V method at different frequencies.

Secondly, the four-probe Hall test could be used to obtain the Hall mobility of IGZO thin films with the same thickness in JFETs.

Thirdly, authors should use the same IGZO process and film thickness as JFET in this work to prepare IGZO MOSFETs on reliable SiO_2 dielectric, and use the calculation method of MOSFETs to obtain the device field effect mobility. By comparing these methods, the mobility extraction can be more credible.

Reply:

We sincerely appreciate the critical insights of the reviewer regarding the mobility estimation of our work. In response, we have carefully addressed each of the reviewer's concerns point by point, as outlined below.

1. Reliability of Hall-effect-based N_s values

We acknowledge the concern of the reviewer that carrier concentration (N_s) extracted from Hall measurements can be affected by the measurement condition or film thickness, potentially causing error in mobility estimation *via* the expression: $\mu = (L/W)(g_m)/(q \cdot N_s \cdot t)$. To address this, we have provided detailed experimental conditions for the Hall measurements in **Revision Figure 1**.

Revision Figure 1. Hall measurement details.

As shown in **Revision Figure 1a**, a rectangular IGZO film with four-point contacts was fabricated following the van der Pauw configuration to ensure reliable sheet resistance (R_s) extraction. The specific dimensions and contact geometry are illustrated in **Revision Figure 1b**. The van der Pauw configuration, a widely accepted method for characterizing uniform thin films, was adopted to minimize geometric error and ensure accurate determination of sheet resistance and Hall voltage. To reduce measurement artifacts due to contact resistance, Al/Au bilayer electrodes were used. Al was employed to form an ohmic contact with the IGZO film, while the Au overlayer served to prevent oxidation of the underlying Al and ensure long-term chemical stability and electrical robustness of the contact. **Revision Figure 1c** shows the AFM-measured thickness of the IGZO layer in the Hall device. Importantly, this film was deposited under identical conditions to those used in the b-JFETs discussed in the main manuscript. This process consistency ensures that the extracted N_s values can be directly applied to the mobility analysis of our devices without additional uncertainty. Taken together, the minimized contact resistance, carefully defined geometry, and identical IGZO film conditions between Hall and JFET devices support the reliability of our Hall-derived carrier concentration values. Moreover, the use of the van der Pauw configuration—an industry-standard technique for extracting carrier properties in thin films—further reinforces the accuracy and general validity of the measurement.

2. Fundamental distinction between JFET and MOSFET mobility extraction

We respectfully clarify that our b-FET device operates via a gate-channel junction mechanism, fundamentally different from the gate dielectric modulation in conventional MOSFETs. Consequently, using gate oxide capacitance C_{OX} in the MOSFET mobility formula is not applicable for evaluating the mobility of our device. Instead, the method employed here is consistent with prior studies on JFET mobility characterization¹⁻⁴.

References

- 1 Lim, J. Y. *et al.* Van der Waals junction field effect transistors with both n- and p-channel transition metal dichalcogenides. *npj 2D Mater. Appl.* **2**, 37 (2018).
- 2 Guo, J. *et al.* SnSe/MoS₂ van der Waals heterostructure junction field-effect transistors with nearly ideal subthreshold slope *Adv. Mater.* **31**, 1902962 (2019).
- 3 Zhu, L. *et al.* Thinning solution-proceed 2D Te for p- and n-channel junction field effect transistor with high mobility and ideal subthreshold slope. *Adv. Funct. Mater.* **34**, 23169488 (2019).
- 4 Chen, X. *et al.* Dual-junction field-effect transistor with ultralow subthreshold swing approaching the theoretical limit. *ACS Appl. Mater. Interfaces.* **16**, 23452-23458 (2024).

3. C-V characterization

We agree with the reviewer's suggestion that calculating carrier mobility using low-frequency capacitance can help avoid overestimation, when using field-effect transistor (FET) models. Accordingly, we performed capacitance-voltage (C-V) measurements of an ITO-IGZO-perovskite-Au test structure across a wide frequency range (0.1 to 100,000 Hz), as shown in Revision Figure 2. The measured capacitance decreases under negative bias to the Au electrode and increases under positive bias (across all frequencies), which is consistent with typical PN junction behavior. Specifically, the capacitance corresponds to depletion (junction) capacitance under reverse bias and diffusion capacitance under forward bias, as described by the relation: $C_j = Ae/W$.

However, we clarify that our device does not utilize a gate dielectric or rely on capacitive charge accumulation, as is the case in conventional FETs. Supplementary Fig. 6 confirms that the perovskite layer in our device behaves as a semiconductor rather than as a gate insulator, and thus forms a rectifying junction with IGZO. Therefore, the capacitance extracted from our measurements cannot be interpreted as the oxide capacitance (C_{OX}) required in traditional FET-based mobility extraction.

Consequently, applying a MOSFET-based mobility model that incorporates C_{OX} would not yield physically meaningful results for our device. Instead, our mobility was extracted using a junction-based model consistent with previously reported methods for JFET mobility analysis¹⁻⁴. This approach appropriately reflects the gate-channel junction modulation that governs current flow in our b-JFET, without invoking gate dielectric capacitance.

Revision Figure 2. C-V measurement of the test device at various frequencies.

4. Bench marking against IGZO MOSFET mobility

To further validate our mobility estimation, we fabricated IGZO MOSFETs on 100 nm SiO₂ substrates using the same IGZO deposition conditions and thickness as in the b-JFET devices. The extracted field-effect mobility is presented in **Revision Figure 3**.

Revision Figure 3. Mobility estimation of SiO₂-based IGZO FET.

The observed higher field-effect mobility in our b-JFET ($29.4 \text{ cm}^2\text{V}^{-1}\text{s}^{-1}$) compared to that in a conventional IGZO FET on SiO₂ ($14.9 \text{ cm}^2\text{V}^{-1}\text{s}^{-1}$) can be attributed to the absence of a gate dielectric interface in the JFET that eliminates interfacial scattering commonly observed at the IGZO/SiO₂ interface. Importantly, both mobility values lie within the reasonable and widely reported range for amorphous IGZO thin films (typically $5\text{--}30 \text{ cm}^2\text{V}^{-1}\text{s}^{-1}$), supporting the validity of our extraction methods and confirming the structural integrity of the IGZO channels in both device types².

Reference

1. Kim, J. Y. *et al.* Advancements of amorphous IGZO-based transistors: materials, processing, and devices. *ACS Appl. Electron. Mater.* **7**, 4703-4728 (2025).

We hope this clarification and additional experimental data appropriately address the concerns of the reviewer regarding mobility extraction.

Reviewer #3 (Remarks to the Author):

The revised work demonstrates a commendable effort to refine and enhance the original draft. The improvements in clarity and depth of analysis are evident, particularly in comparison of other literature on Sn-based perovskite and IGZO transistors and clarification of the function of a natural oxidation layer on Sn perovskites. In general, this revision marks significant progress toward achieving the work's objectives.

-----”-----
In a confidential remark to the editor, Reviewer #3 also finds the claim of low energy consumption of the perovskite/IGZO device comparing to pure IGZO-device is not supported by any data, so requests for the evaluation of the power consumption.

Reply:

We appreciate the comment of the reviewer regarding the power consumption of our device. As noted in our response to the comment of Reviewer 2, the static power consumption (P_{static}) of our inverter can be estimated using the relation: $P_{\text{static}} = V_{\text{DD}} \times I_{\text{static}}$. In our NMOS inverter, static power primarily arises from leakage current through the pull-up JFET when it is in the off state (i.e., $V_{\text{GS}} = 0$ V). From the transfer curve, the drain current under this condition is approximately 6.6×10^{-10} A. Given $V_{\text{DD}} = 1$ V, the corresponding P_{static} is approximately 660 pW. This remarkably low static power is attributed to the enhancement mode operation of the b-JFET, ensuring near-zero current flow in the standby state. This point has been emphasized in the revised main text.

Page 13, line 2: In addition, the inverter exhibited extremely low static power consumption of approximately 660 pW at $V_{\text{DD}} = 1$ V, owing to the enhancement mode operation of the b-JFET that ensures negligible standby current ($I_{\text{static}} \approx 6.6 \times 10^{-10}$ A when $V_{\text{IN}} = 0$ V).